# Kramers nodal line metals

Ying-Ming Xie [1,3], Xue-Jian Gao [1,3], Xiao Yan Xu [2], Cheng-Ping Zhang[1], Jin-Xin Hu[1], Jason Z. Gao[1] & K. T. Law [1✉]

Recently, it was pointed out that all chiral crystals with spin-orbit coupling (SOC) can be Kramers Weyl semimetals (KWSs) which possess Weyl points pinned at time-reversal invariant momenta. In this work, we show that all achiral non-centrosymmetric materials with SOC can be a new class of topological materials, which we term Kramers nodal line metals (KNLMs). In KNLMs, there are doubly degenerate lines, which we call Kramers nodal lines (KNLs), connecting time-reversal invariant momenta. The KNLs create two types of Fermi surfaces, namely, the spindle torus type and the octdong type. Interestingly, all the electrons on octdong Fermi surfaces are described by two-dimensional massless Dirac Hamiltonians. These materials support quantized optical conductance in thin films. We further show that KNLMs can be regarded as parent states of KWSs. Therefore, we conclude that all non-centrosymmetric metals with SOC are topological, as they can be either KWSs or KNLMs.

[1] Department of Physics, Hong Kong University of Science and Technology, Hong Kong, China. [2] Department of Physics, University of California at San Diego, La Jolla, CA, USA. [3]These authors contributed equally: Ying-Ming Xie, Xue-Jian Gao. ✉email: phlaw@ust.hk

The discovery of topological insulators[1–7] that possess bulk insulating gap and massless Dirac surface states have inspired intense theoretical and experimental studies in the symmetry and topological properties of electronic band structures. In recent years, a large number of topological insulators and topological semimetals, such as topological crystalline insulators[8], higher-order topological insulators[9–13], Dirac semimetals[14–24], Weyl semimetals[25–37], nodal line[38–42], nodal chain[43], and multifold chiral[44–53] topological semimetals, have been discovered. Moreover, systematic ways to diagnose non-trivial band topology based on topological quantum chemistry and symmetry-based indicators have been developed and a large number of topological materials have been found[54–58].

Recently, the study of Kramers Weyl semimetals (KWSs) has significantly expanded the family of topological materials[59]. It has been stated that in all chiral crystals (crystals that lack mirror or roto-inversion symmetries) with spin–orbit coupling (SOC), each twofold degenerate time-reversal invariant momentum (TRIM) point is a Weyl point called Kramers Weyl point. Around a Kramers Weyl point, the degeneracy near the TRIM is split along all directions in momentum space by SOC[60]. Consequently, the Fermi pockets enclosing Kramers Weyl points are split by SOC, and each Fermi pocket possesses nontrivial and opposite Chern numbers, as depicted in Fig. 1a[59]. These KWSs exhibit several novel properties, such as the monopole-like spin texture[59,61], longitudinal magnetoelectric responses[62,63], and the quantized circular photogalvanic effect[52,59,64–67].

In this work, we point out that all non-centrosymmetric achiral crystals (crystals that possess mirror or roto-inversion symmetries) with SOC possess doubly degenerate lines, which connect TRIM points with achiral little group symmetry across the Brillouin zone. The double degeneracy is protected by time-reversal and achiral point group symmetries of the crystal. We call these doubly degenerate lines, Kramers nodal lines (KNLs). It is shown that these KNLs exist in all non-centrosymmetric achiral crystals with SOC. When the Fermi surfaces of materials enclose TRIM points connected by KNLs, we call these materials Kramers nodal

lines metals (KNLMs). In Table 1, all the symmorphic space groups (SGs) supporting KNLs are listed, and certain material realizations are identified.

Importantly, as long as the Fermi surfaces enclose TRIMs that are connected by KNLs, the KNLs force spin-split Fermi surfaces to touch on the KNLs and create two types of Fermi surfaces, namely, the spindle torus type and the octdong (or hourglass) type as shown in Fig. 1b, d, respectively. The band touching points of the Fermi surfaces are described by two-dimensional massless Dirac or higher-order Dirac Hamiltonians[20,50,68,69], with the Dirac points pinned at the Fermi energy. In the case of octdong-type Fermi surfaces, all the states on the Fermi surfaces are described as two-dimensional massless Dirac fermions. Materials with octdong-type Fermi surfaces exhibit linear optical conductivity in the bulk and, in the thin film limit, quantized optical conductivity similar to monolayer graphene due to the massless Dirac fermions[70,71].

Furthermore, KNLMs can be regarded as the parent states of KWSs. When the mirror or roto-inversion symmetries are broken, the degeneracies of the KNLs are lifted, and the touching points of the Fermi surface will generally be gapped out and a KNLM becomes a KWS. More specifically, breaking achiral crystal symmetries causes a spindle Fermi surface (Fig. 1b) to split into two Fermi pockets as shown in Fig. 1a, and each Fermi pocket carries a net Chern number. In the case of an octdong Fermi surface (Fig. 1b), the two Fermi pockets detach from each other and Kramers Weyl points are generated in both pockets, as shown in Fig. 1c. For illustration, we demonstrate how an isolated Kramers Weyl point near the Fermi energy can be created by breaking the mirror symmetry through strain in BiTeI with a spindle Fermi surface, and how this Kramers Weyl point can be detected through the quantized circular photogalvanic effect[64].

From this work, together with the discovery of KWSs, we conclude that all non-centrosymmetric crystals with SOC are topological in nature. They can be either KWSs or KNLMs.

## Results

**Emergence of Kramers nodal lines from TRIMs with achiral little group symmetry.** In this section, we demonstrate how nodal lines emerge out of a TRIM with achiral little group symmetry (which contains mirror or roto-inversion). According to Kramers theorem, each electronic band is at least doubly degenerate at a TRIM $\mathbf{k}_0$, where $\mathbf{k}_0 = -\mathbf{k}_0 + \mathbf{G}_i$, and $\mathbf{G}_i$ denotes a reciprocal lattice vector. We first focus on the cases that the energy bands are twofold degenerate at TRIM points, and the cases with fourfold degeneracy are discussed in the "Methods" section. In general, the energy bands near the TRIM $\mathbf{k}_0$ with little group symmetry (Supplementary Note 2) $\mathcal{G}_{\mathbf{k}_0}$ can be described by a Hamiltonian

$$H(\mathbf{k}) = f_0(\mathbf{k}) + \boldsymbol{f}(\mathbf{k}) \cdot \sigma, \quad (1)$$

where $\mathbf{k}$ is measured from the TRIM $\mathbf{k}_0$, $\sigma$ are Pauli matrices operating on the spin space, $\boldsymbol{f}(\mathbf{k}) \cdot \sigma$ denotes the SOC and the eigenvalues of $H(\mathbf{k})$ can be written as $E \pm (\mathbf{k}) = f_0(\mathbf{k}) \pm |\boldsymbol{f}(\mathbf{k})|$.

As $H(\mathbf{k})$ respects the time-reversal symmetry $\mathcal{T} = i\sigma_y K$ ($K$ is the complex conjugate operation) and the little group symmetry $\mathcal{G}_{\mathbf{k}_0}$, $\boldsymbol{f}(\mathbf{k})$ satisfies the symmetry constraints

$$\boldsymbol{f}(\mathbf{k}) = -\boldsymbol{f}(-\mathbf{k}), \boldsymbol{f}(\mathbf{k}) = \mathrm{Det}\,(R)R^{-1}\boldsymbol{f}(R\mathbf{k}), \quad (2)$$

where $R \in \mathcal{G}_{\mathbf{k}_0}$.

For illustration, we analyze the case where $\boldsymbol{f}(\mathbf{k})$ is linear in $\mathbf{k}$, i.e., $\boldsymbol{f}(\mathbf{k}) = \hat{M}\mathbf{k}$, where $\hat{M}$ is a matrix. A more general proof is provided in the Supplementary Note 2. According to Eq. (2), $\hat{M}$ satisfies $\hat{M} = \mathrm{Det}\,(R)R^{-1}\hat{M}R$. Denoting $\mathbf{n}_j$ and $\epsilon_j$ as the eigenstates and the eigenvalues of matrices $\hat{M}$ satisfying

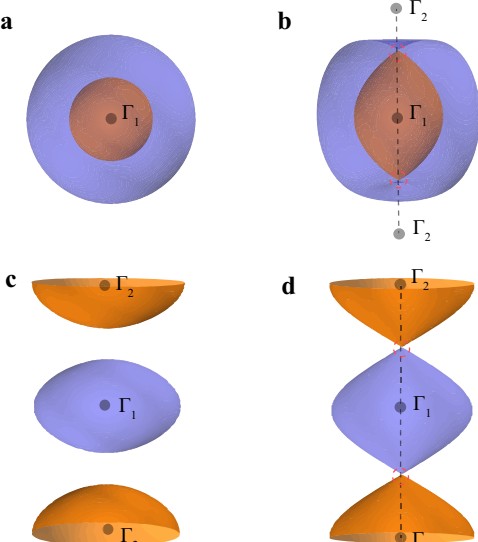

**Fig. 1 Schematic plot of Fermi surfaces of KWSs and KNLMs. a** The Fermi surface of a KWS where two Fermi pockets enclose one TRIM. **b** Spindle torus-type Fermi surface in a KNLM induced by a KNL (the dashed black line). **c** The Fermi surface of a KWS where each pocket encloses a different TRIM. **d** Octdong-type Fermi surface in KNLMs induced by a KNL. The gray dots in **a–d** indicate the position of TRIMs $\Gamma_1$, $\Gamma_2$. The touching points of the Fermi surfaces are circled by red dashed lines.

**Table 1 Kramers nodal line metals (KNLMs) with symmorphic space groups[a].**

| Type | SG no. | Point group | KNLs | KW points | Material |
|---|---|---|---|---|---|
| Type I | 6, $Pm$ | $C_{1v}$ | $(\Gamma, B, Y, A, Z, C, D, E)$[b] | – | $CsIO_3$ |
| | 8, $Cm$ | $C_{1v}$ | $(\Gamma, Y, A, M)$ | – | $BiPd_2Pb$ |
| | 25, $Pmm2$ | $C_{2v}$ | $\Gamma$–Z, Y–T, X–U, S–R | – | $CdTe$, $Bi_4Te_2Br_2O_9$ |
| | 38, $Amm2$ | $C_{2v}$ | $\Gamma$–Y, T–Z | – | $NbS_2$ |
| | 42, $Fmm2$ | $C_{2v}$ | $\Gamma$–Z, Y–T | – | – |
| | 99, $P4mm$ | $C_{4v}$ | $\Gamma$–Z, X–R, A–M | – | $PbCsCl_3$ |
| | 107, $I4mm$ | $C_{4v}$ | $\Gamma$–M, X–X, (N) | – | $In_2Te_3$ |
| | 115, $P\bar{4}m2$ | $D_{2d}$ | $\Gamma$–Z, M–A, X–R | – | $PbF_2O$ |
| | 156, $P3m1$ | $C_{3v}$ | $\Gamma$–A, (M, L) | – | $BiTeI$ |
| | 157, $P31m$ | $C_{3v}$ | $\Gamma$–A, (M, L) | – | $Bi_2Pt$ |
| | 160, $R3m$ | $C_{3v}$ | $\Gamma$–T, (L, FA) | – | $Bi_2Te_3$ |
| | 174, $P\bar{6}$ | $C_{3h}$ | $\Gamma$–A, (M, L) | – | – |
| | 183, $P6mm$ | $C_{6v}$ | $\Gamma$–A, M–L | – | $AuCN$ |
| | 187, $P\bar{6}m2$ | $D_{3h}$ | $\Gamma$–M, A–L, $\Gamma$–A | – | $GeI_2$, $TaN$ |
| | 189, $P\bar{6}2m$ | $D_{3h}$ | $\Gamma$–K–M, A–H–L, $\Gamma$–A | – | $Sn_5(BIr_3)_2$ |
| | 215, $P\bar{4}3m$ | $T_d$ | $\Gamma$–X, $\Gamma$–R, R–M | – | $Cu_3TaTe_4$ |
| | 216, $F\bar{4}3m$ | $T_d$ | $\Gamma$–L, $\Gamma$–X | – | $HgSe$, $HgTe$ |
| | 217, $I\bar{4}3m$ | $T_d$ | $\Gamma$–H | – | $TaTl_3Se_4$ |
| Type II | 35, $Cmm2$ | $C_{2v}$ | $\Gamma$–Z, Y–T | S, R | $MnCs_2V_2Br_2O_6$ |
| | 44, $Imm2$ | $C_{2v}$ | $\Gamma$–X, (S, R) | T | $AgNO_2$ |
| | 81, $P\bar{4}$ | $S_4$ | $\Gamma$–Z, M–A | X, R | $GeSe_2$ |
| | 82, $I\bar{4}$ | $S_4$ | $\Gamma$–M | N, X | $CdGa_2Te_4$, $Cr_2AgBiO_8$ |
| | 111, $P\bar{4}2m$ | $D_{2d}$ | $\Gamma$–Z, M–A | X, R | $Ag_2HgI_4$ |
| | 119, $I\bar{4}m2$ | $D_{2d}$ | $\Gamma$–M, (N) | X | $TlAgTe_2$ |
| | 121, $I\bar{4}2m$ | $D_{2d}$ | $\Gamma$–M, X–X | N | $Cu_3SbS_4$ |

[a]Here, we enumerate symmetry allowed KNLs in symmorphic space groups. The definitions of TRIMs follow the conventions given in Bilbao Crystallographic Server[73]. Some of the representative materials hosting KNLs are identified with the assistance of the Materials Project[90] and the Topological Material Database[58].
[b]The TRIMs in the parentheses are connected by the KNLs that are not along the high symmetry lines, such as $(\Gamma, A)$, $(Y, M)$ in SG no. 8 ($Pm$) and $(M, L)$ in SG no. 156 ($P3m1$).

$\hat{M}\mathbf{n}_j = \epsilon_j \mathbf{n}_j$, and decomposing the momentum $\mathbf{k}$ with the new basis as $\mathbf{k} = \sum_j p_j \mathbf{n}_j$, one finds

$$f(\mathbf{k}) = \sum_j p_j \epsilon_j \mathbf{n}_j. \tag{3}$$

In general, for a TRIM with a little group symmetry that is chiral, $\text{Det}(\hat{M}) \neq 0$, namely $\epsilon_j$ are all finite. In this case, $|f(\mathbf{k})| > 0$ as long as $\mathbf{k}$ is not at the TRIM, which results in a fully split Fermi surface as shown in Fig. 1a and makes the TRIM a Kramers Weyl point as pointed out in ref. [59]. In contrast, for a TRIM with an achiral little group, there exists at least one mirror or roto-inversion operation $\tilde{R}$ with $\text{Det}(\tilde{R}) = -1$ such that $\text{Det}(\hat{M}) = 0$, implying that at least one of $\epsilon_j$ is zero. Without loss of generality, taking $\epsilon_3 = 0$, one obtains

$$f(\mathbf{k}) = p_1 \epsilon_1 \mathbf{n}_1 + p_2 \epsilon_2 \mathbf{n}_2. \tag{4}$$

$f(\mathbf{k})$ vanishes when the momentum $\mathbf{k}$ is fixed to be along the direction of null vector $\mathbf{n}_3$, where $p_1 = p_2 = 0$ and $\mathbf{k} = p_3 \mathbf{n}_3$. In this case, $E_+(\mathbf{k})$ and $E_-(\mathbf{k})$ are degenerate along the $\mathbf{n}_3$-direction. The line $\mathbf{k} = p_3 \mathbf{n}_3$ is an example of a degenerate line coming out of TRIMs. The degeneracy is protected by time-reversal symmetry and the achiral little group symmetry. We called these lines, KNLs. It is important to note that KNLs create touching points on the Fermi surface at any Fermi energy as long as the Fermi surface enclose TRIMs with achiral little groups, as depicted schematically in Fig. 1. Interestingly, these touching points, which are always pinned at the Fermi energy, are two-dimensional Dirac points or higher-order Dirac points[20,50,68,69] with nontrivial topological properties (Supplementary Note 3). The general form of the $\mathbf{k} \cdot \mathbf{p}$ Hamiltonians of all non-centrosymmetric achiral point groups and the directions of KNLs, emerging out of the TRIM are summarized in the "Methods" section. Beyond the $\mathbf{k} \cdot \mathbf{p}$ analysis, we showed in the

Supplementary Note 2 that for a general $f(\mathbf{k})$, the KNLs are guaranteed to lie within the mirror planes or along the roto-inversion axis of $S_3$, $S_4$ symmetry. It is further shown that a KNL emerging from one TRIM has to connect with another TRIM, with an achiral little group (Supplementary Note 2).

**Kramers nodal lines in achiral crystals.** In the previous section, we demonstrated how KNLs emerge out of TRIMs. In this section, we study how KNLs connect different TRIMs in non-centrosymmetric achiral crystals. While most KNLs connect TRIMs along high symmetry lines, some KNLs connect TRIMs through general points in the mirror plane (such as for TRIMs with $C_{1v}$ little groups).

To identify the KNLs joining TRIMs along high symmetry lines, we make use of the compatibility relations of double-valued SGs[72,73], which are defined by

$$\chi\left(D_{\mathcal{G}_1}^{(\Gamma_1)}(R)\right) = \sum_j \chi\left(D_{\mathcal{G}_2}^{(\Gamma_j)}(R)\right), \tag{5}$$

where $\chi$ is the character of a symmetry operation $R$ in a specific representation, $\mathcal{G}_1$ and $\mathcal{G}_2$ are the little groups of the TRIM and a high symmetry line, respectively, and $D_{\mathcal{G}_i}^{(\Gamma_j)}(R)$ is the $j$th irreducible representation of the symmetry operation $R \in \mathcal{G}_i$. For example, for the well-studied 3D Rashba material BiTeI (SG no. 156, $P3m1$), the little groups of the TRIM $\Gamma$, A and the high symmetry line $\Delta$ connecting these two TRIMs are all $C_{3v}$. By identifying the irreducible representations of the unitary symmetry operations $m_{010}$ and $C_3$ at $\Gamma$, A, and $\Delta$ (see Supplementary Note 4 for details), we show that the two-dimensional double-valued irreducible representations $\bar{\Gamma}_6$–$\bar{\Delta}_6$–$\bar{A}_6$ are compatible. This explains all the KNLs $\Gamma$–A observed in the band structure of BiTeI shown in Fig. 2c (labeled with blue color). This result is also consistent with

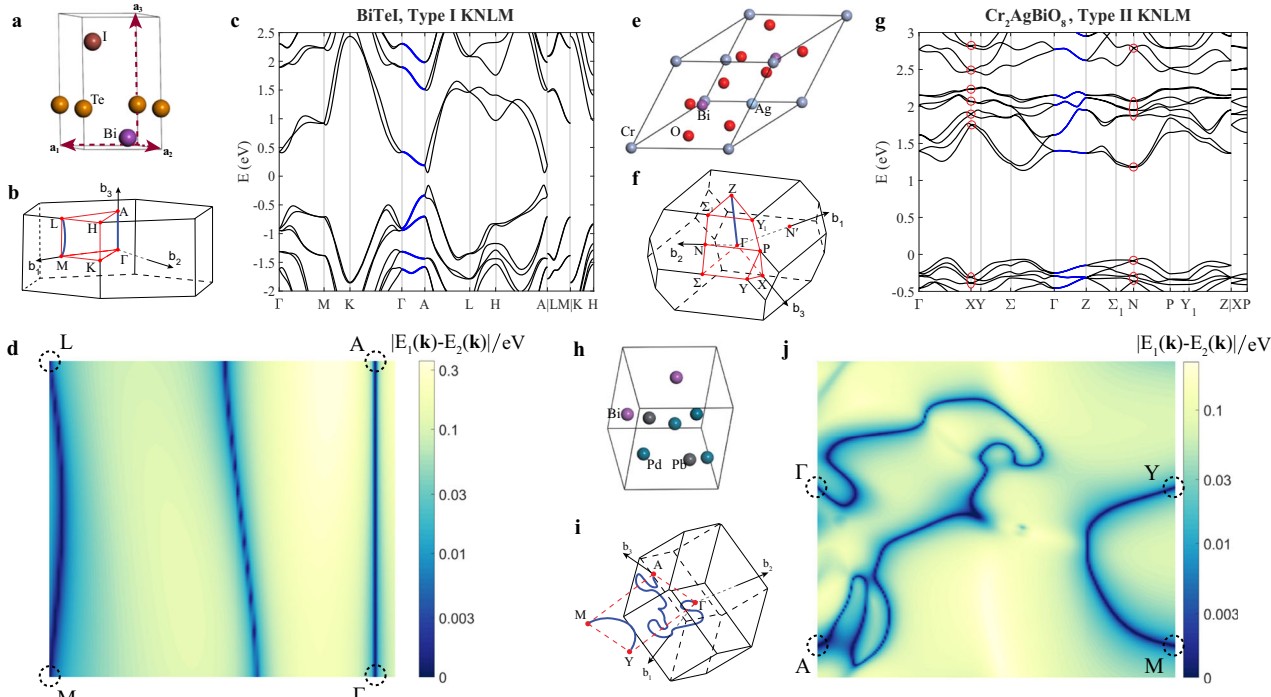

**Fig. 2 Representative materials with KNLs. a–j** The crystal structure, the first Brillouin zone, and KNLs of BiTeI (SG no. 156, $P3m1$), $Cr_2AgBiO_8$ (SG no. 82, $I\bar{4}$), and $BiPd_2Pb$ (SG no. 8, $Cm$). **c, g** The band structures of BiTeI and $Cr_2AgBiO_8$, respectively, where the KNLs are highlighted as blue lines, and the crossing points within the red circles of **f** are KW points. These KNLs are also marked out by solid blue lines in the 3D first Brillouin zone. **d, j** The DFT-calculated energy difference of two selected SOC-split bands $|E_1(\mathbf{k}) - E_2(\mathbf{k})|$ (in units of eV) on a mirror-invariant $k$ plane for BiTeI and $BiPd_2Pb$, respectively. The dark green lines that connect two TRIMs (dashed circles) are KNLs on this mirror plane.

the $\mathbf{k} \cdot \mathbf{p}$ Hamiltonian analysis that a KNL emerges out of the $\Gamma$ point along the $z$-direction (see the "Methods" section).

Based on the compatibility relations, we identified all the KNLs that are along the high symmetry lines in non-centrosymmetric crystals with symmorphic SGs. The results are summarized in Table 1. We found non-centrosymmetric achiral crystals with point groups $C_{2v}$, $S_4$, $C_{4v}$, $D_{2d}$, $C_{3v}$, $C_{3h}$, $C_{6v}$, $D_{3h}$, $T_d$ support KNLs along high symmetry directions. These lines are contained within the mirror plane or along the roto-inversion axis. Some representative materials with KNLs are listed in Table 1. For example, for SG 216, there are KNLs along the high symmetry lines between $\Gamma$ and L points, as well as between $\Gamma$ and X points. These KNLs are labeled as $\Gamma$–L and $\Gamma$–X, respectively, in Table 1. Materials with this property include semimetals HgTe and HgSe. For further illustration, the band structures of BiTeI (SG no. 156, $P3m1$) and $Cr_2AgBiO_8$ (SG no. 82, $I\bar{4}$) are shown in Fig. 2. Evidently, there are KNLs (labeled with blue color) along the high symmetry lines.

Although most KNLs reside on high symmetry lines, there are exceptions if the little group of the TRIM is $C_{1v}$. As shown in the previous section, $C_{1v}$ is achiral so that there must be KNLs emerging from TRIMs. For example, the little groups of TRIMs M and L in BiTeI are the achiral $C_{1v}$, yet there are no KNLs along high symmetry lines coming out from M or L, as shown in Table 1. However, by carefully checking the energy bands on the whole mirror plane, as shown in Fig. 2d (and schematically shown in Fig. 2b), we indeed found a KNL that connects M, L within the mirror plane that is denoted as (M, L) in Table 1. Therefore, all TRIMs in BiTeI are connected by KNLs as expected.

On the other hand, there exist TRIMs with chiral little group symmetry, such as the X and N points in achiral KNLM $Cr_2AgBiO_8$. Therefore, the Bloch states for each band near X and

N points in $Cr_2AgBiO_8$ are described by Kramers Weyl fermions, as highlighted in Fig. 2g. As demonstrated in the Supplementary Note 7, Fermi arcs originating from these Kramers Weyl points emerge on (001) surfaces of $Cr_2AgBiO_8$. As summarized in Table 1, among the 25 non-centrosymmetric achiral symmorphic SGs, 18 of them are classified as type I achiral crystals, in which all the TRIMs are connected by KNLs. In contrast, the remaining seven SGs further support Kramers Weyl points, and they are classified as type II achiral crystals.

One interesting example of KNLs can be found in $BiPd_2Pb$ (SG no. 8, $Cm$, point group $C_{1v}$), which exhibits large SOC-induced band splitting ~100 meV (see Supplementary Note 7 for the band structure). The lattice structure and the Brillouin zone is shown in Fig. 2h, i, respectively. In Fig. 2j, we select two bands that are degenerate on the TRIMs and plot the energy difference with respect to momentum $\mathbf{k}$ in the mirror plane (see the detail band structure in Supplementary Note 7). Remarkably, there are two KNLs, ($\Gamma$–A) and (Y–M), lying on this mirror plane as expected. The schematic plot of the KNLs on the mirror plane is depicted in Fig. 2i. While KNLs along high symmetry lines can easily be found in standard band structure calculations, this kind of irregular KNLs coming out of TRIM with $C_{1v}$ little groups can easily be missed.

**Spindle torus-type and octdong-type Fermi surfaces**. In this section, we point out an important physical consequence of the KNLs, namely, KNLs force SOC-split Fermi surface to touch. Interestingly, there are two kinds of Fermi surface touchings, which can satisfy the doubly degenerate requirement of KNLs. The first type is the spindle torus Fermi surface formed by the touching of two electron Fermi pockets, as illustrated schematically in Fig. 1b, in which the KNL forces the two SOC split Fermi

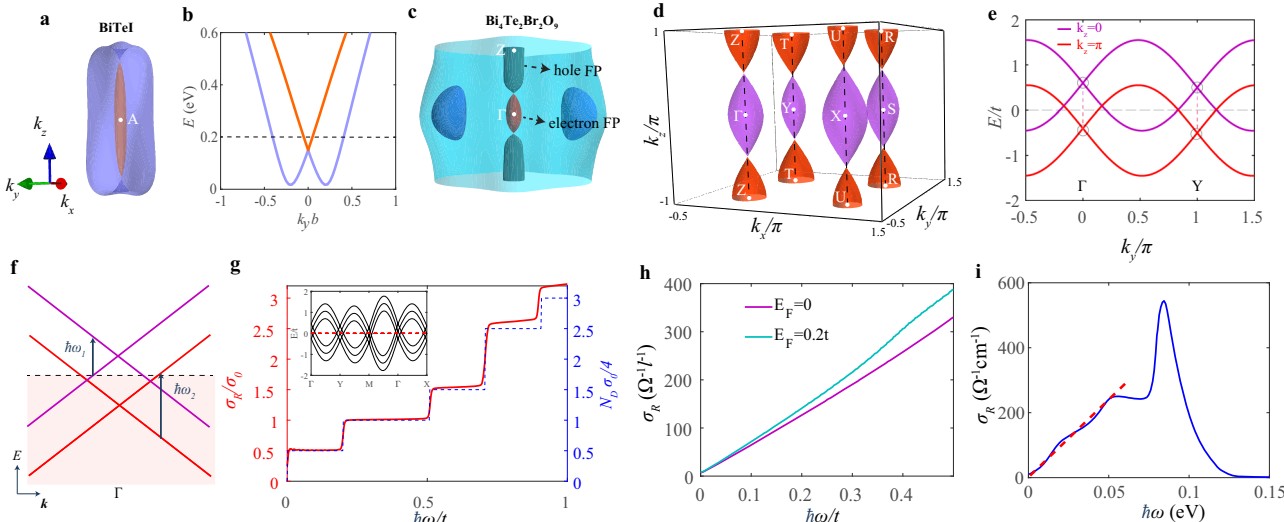

**Fig. 3 Spindle torus and octdong Fermi surfaces. a** The Fermi surface of BiTeI with Fermi energy $E_F = 0.2$ eV, which cuts through the KNL Γ–A. The inner (orange) and outer (purple) Fermi pockets (FP) together form a spindle torus. The energy dispersion at a fixed $k_z$ indicated by the dashed line is shown in **b**. **b** The Rashba-like energy dispersion for a fixed $k_z$. **c** The Fermi surface of $Bi_4Te_2Br_2O_9$ (SG no. 25, *Pmm*2) with Fermi energy $E_F = 0.05$ eV, which cuts through the KNL Γ–Z. The labeled hole and the electron Fermi pockets together form an octdong-type Fermi surface. **d** The Fermi surface from the two-band tight-binding model $\mathcal{H}_0(\mathbf{k})$ with $m_x = 0.05t$, $m_y = 0.05t$, $m_z = 0.5t$, $v_x = t$, $v_y = t$, $E_F = 0$, and $t = 1$ as the unit of the hybridization energy. The positions of TRIMs depicted are all connected by four KNLs in the $k_z$-direction. **e** The energy dispersion for a fixed $k_z = 0$ (purple) and $k_z = \pi$ (red) in **d**. **f** Schematic plot of optical excitations that contribute to the optical conductivity for the hole-type (electron-type) Dirac fermions with onset frequency $\omega_1$ ($\omega_2$). The horizontal dashed line denotes the position of Fermi energy. **g** The optical conductivity $\sigma_R$ (left axis) and estimated optical conductivity $N_D\sigma_0/4$ (right axis) versus frequency $\omega$ for a three-layer slab, where the number of Dirac points $N_D = \frac{1}{2}\sum_{\Gamma,n}\theta(\hbar\omega - |2E_{\Gamma,n}|)$ with $\theta$ as the Heaviside step function, $n$ as band index, and Γ labeling four TRIMs. The inset figure in **g** shows the band structure of this trilayer slab. **h** The bulk optical conductivity for the model material with octdong Fermi surface at $E_F = 0$, $0.2t$ with $\eta = 0.002t$ and temperature $T = 0.01t$. Here, $l^{-1} = \frac{2\pi}{\bar{a}}$ cm$^{-1}$ with $\bar{a} = a/$Å and $a$ as the lattice constant. **i** The bulk optical conductivity for $Bi_4Te_2Br_2O_9$ with $\eta = 1$ meV and temperature $T = 10$ K. The slight deviation from linear dependence (red dashed line) for $Bi_4Te_2Br_2O_9$ is due to the presence of the extra trivial pockets (blue pockets in **c**).

pockets to touch. The spindle torus Fermi surfaces are rather common in achiral crystals with strong SOC. It is well-known that BiTeI possesses this kind of Fermi surface[74], and we explain here that the origin of the Fermi surface touching is indeed enforced by the Γ–A KNL, as illustrated in Fig. 3a. To understand the properties of the electrons on spindle Fermi surfaces, we use BiTeI as an example and note that with a fixed $k_z$, the electrons on the Fermi surface are described by a two-dimensional Rashba Hamiltonian as illustrated in Fig. 3b[75,76]. In this work, we point out that almost all non-centrosymmetric achiral crystals with strong SOC have similar properties even though the Fermi surfaces can be more complicated. In the case of hole-doped HgTe and HeSe, for example, three KNLs come out of the Γ point and result in six Fermi surface touching points, as illustrated in the Supplementary Note 7.

The second type of Fermi surface touchings that satisfies the degeneracy requirement on the KNLs is the octdong-type Fermi surface. In this case, one electron Fermi pocket and one hole Fermi pocket touch along the KNL, as illustrated in Fig. 1b schematically and in Fig. 3c, using the realistic band structures of $Bi_4Te_2Br_2O_9$ (SG no. 25, *Pmm*2, point group $C_{2v}$). In $Bi_4Te_2Br_2O_9$, there is an octdong Fermi surface near the Γ point, and the KNL is along the Γ–Z direction. It is important to note that this Fermi surface touching is not accidental, but forced by the KNL. As the chemical potential changes, the relative size of the electron and hole pockets changes and the band touching point moves along the KNL. Importantly, for a fixed $k_z$ along the nodal line direction, the electrons on the octdong Fermi surface are described by two-dimensional massless Dirac fermions on the whole Fermi surface.

The octdong Fermi surface as well as the trivial Fermi sheet of $Bi_4Te_2Br_2O_9$ in Fig. 3c can be captured by a simple tight-binding Hamiltonian, which satisfies the SG symmetry SG no. 25 (*Pmm*2). The effective Hamiltonian can be written as

$$\mathcal{H}_0(\mathbf{k}) = \sum_j m_j \cos(k_j) + v_x \sin k_x \sigma_x + v_y \sin k_y \sigma_y, \quad (6)$$

where $j = x$, $y$, $z$, $\sigma$ are Pauli spin matrices. As illustrated in Fig. 3d, it is interesting to note that symmetry allows the crystal to possess pure octdong Fermi surfaces, when SOC is further enhanced. Unfortunately, we have yet to identify realistic materials with pure octdong Fermi surfaces.

To understand the novel properties of octdong Fermi surfaces, we first study the optical properties of a system with the octdong Fermi surface only as depicted in Fig. 3d. The cases with additional trivial Fermi surfaces will be discussed later. We note that in the case of Fig. 3d, all the electrons on the Fermi surface are described by two-dimensional massless Dirac fermions with Dirac points located on the KNLs. The massless Dirac energy dispersions at $k_z = 0$ and $k_z = \pi$ are depicted in Fig. 3e. It is clear from Fig. 3e that the energy bands cross at Γ and Y points, which are Dirac points. Dirac points corresponding to general $k_z$ lie along the dashed lines in Fig. 3e between the two Dirac points highlighted by circles. In other words, all the states on the octdong Fermi surface can be described by two-dimensional massless Dirac Hamiltonians, and the energy of the Dirac points is determined by $k_z$. We expect the large number of Dirac electrons on octdong surfaces possess novel physical properties.

To illustrate this, we calculate the optical conductivity $\sigma_R(\omega) \equiv \text{Re}(\sigma_{xx}(\omega))$ for a thin film of material with the octdong Fermi

surface, using a tight-binding version of the effective Hamiltonian (Eq. (6)). The energy spectrum of such a trilayer thin film is shown in the insert of Fig. 3g, which can be effectively described by multiple massless Dirac Hamiltonians. Applying the Kubo formula, the optical conductivity can be written as

$$\sigma_R(\omega) = \frac{e^2}{\hbar V} \sum_{\mathbf{k}} \sum_{i \neq j} \frac{f(\epsilon_i(\mathbf{k})) - f(\epsilon_j(\mathbf{k}))}{\epsilon_i(\mathbf{k}) - \epsilon_j(\mathbf{k})} \cdot$$
$$|\langle i, \mathbf{k}|\hat{v}_x|j, \mathbf{k}\rangle|^2 \operatorname{Im}\left( \frac{1}{\hbar\omega + i\eta + \epsilon_i(\mathbf{k}) - \epsilon_j(\mathbf{k})} \right),$$
$$(7)$$

where $\omega$ is the frequency of the incident light, $V$ is the volume (area) for a bulk (thin film) sample, $i, j$ are the band indices, $f$ is the Fermi–Dirac distribution function, $\eta$ originating from the effect of carrier damping is assumed to be a constant, and $\hat{v}_x = \partial\mathcal{H}_0/\partial k_x$ is the velocity operator. As shown in Fig. 3g, remarkably, the optical conductivity is quantized and shows plateau structures. The quantization is similar to monolayer graphene that exhibits quantized optical conductivity of $\sigma_0 = \pi e^2/2h$ in the frequency range $\omega > 2|\mu|$, with $\mu$ being the chemical potential measured from the Dirac point[70,77,78]. To understand the plateau structure, we note that different Dirac points of the thin film have different activation frequencies at which light can excite occupied states into empty states, as depicted in Fig. 3f. As the optical frequency increases, more and more optically activated Dirac points contribute to quantized optical conductivity and result in the plateau structure. By counting the number of Dirac points $N_D$ within half of the optical frequency $\omega$, we obtain the quantized plateaus (blue dashed line in Fig. 3g) that is consistent with the one calculated with the Kubo formula (Eq. (7)). This clearly demonstrates the novel properties of materials with octdong Fermi surfaces. The deviation from the quantization values at higher frequencies is due to the deviation from the Dirac energy spectrum at energy far from the Dirac points.

The number of two-dimensional massless Dirac fermions are expected to scale with the system size. In the bulk limit, the optical conductivity with octdong Fermi surfaces is linearly proportional to the optical frequency due to the large number of two-dimensional massless Dirac fermions, as denoted by the linear line in Fig. 3h. Importantly, the onset frequency for this linear line is pinned at zero, regardless of chemical potential (Fig. 3h). The underlying reason is that those touching points on the octdong Fermi surface always manifest as massless Dirac points right at Fermi energy. This is substantially different from the linear optical conductivity shown in Weyl[79,80], Dirac semimetals[81–83], and multi-fermions[84], where the onset frequency depends on how far the chemical potential is away from the Weyl or Dirac points. Moreover, as shown in Fig. 3i, in the case of the coexistence of an octdong Fermi surface and trivial

Fermi surfaces in $Bi_2Te_2Br_2O_9$, the optical conductivity, which is calculated from realistic tight-binding models constructed with Wannier orbitals from DFT calculations (Supplementary Note 7), also shows such linear increase, although it is limited to a relatively smaller frequency range. When the optical frequency is high, transitions appear between states that are far from the Dirac points, and the linear behavior of the optical conductivity is lost. To experimentally demonstrate this linear optical conductivity in KNLMs, the incident direction of light should be parallel to the KNLs, and the Drude response that gives a peak near-zero frequency needs to be subtracted[85].

**KNLMs as the parent states of Kramers Weyl materials.** In this section, we point out that KNLMs are parent states of KWSs, and one can obtain KWSs from KNLMs through lattice symmetry breaking. To understand the relation between KNLMs and KWSs, we note that the KNLs are doubly degenerate lines connecting TRIMs. A plane in the Brillouin zone intercepting a KNL can be described by a two-dimensional massless Dirac Hamiltonian with Berry curvature concentrated at the Dirac point. When a Bloch electron moves around a KNL adiabatically, it acquires a quantized Berry phase of $m\pi \bmod 2\pi$ (Supplementary Note 3), and one can regard a KNL carrying Berry curvature flux of $\pi$ as a Dirac solenoid, as illustrated in Fig. 4a. It is important to note that the Berry curvature on the opposite sides of a TRIM should have opposite signs because of time-reversal symmetry such that the Dirac solenoids[86] manifested by KNLs do not have classical analogs. When the symmetries (such as the mirror or the roto-inversion) of a crystal are broken, the degeneracy of the KNLs is lifted, and it is possible to define a nondegenerate Fermi surface enclosing a TRIM. As depicted in Fig. 4b, the Berry flux coming out of a TRIM is quantized. Therefore, the nondegenerate Fermi surface enclosing a TRIM has a finite Chern number on each pocket and the TRIM becomes a Kramers Weyl point.

For illustration, we apply strain on BiTeI to break all the mirror symmetries of the crystal. The compressive strain is achieved by reducing the lattice constant $\mathbf{a_1}$ of the crystal as shown in Fig. 2a. The evolution for the band structures along $\Gamma$–$A$ under 1%, 3%, and 5% strain strengths is summarized in Fig. 4c–e, respectively. (Note that the KNL $\Gamma$–$A$ in the case without strain is shown in Fig. 2c.) Impressively, we found the KNL $\Gamma$–$A$ in BiTeI can be split sizably (~ order of tens of meV) by <3% strain, and the $\Gamma$ and $A$ points become Kramers Weyl points with opposite chirality. As $A$ is the only Weyl point that is close to the Fermi energy, while other Weyl points are at least 200 meV above, a single Weyl point near the Fermi energy is generated. Although there is only a single Weyl point near the Fermi energy, the Nielsen–Ninomiya theorem is not violated because there are two Fermi pockets carrying opposite chiral charges, which enclose this Weyl point.

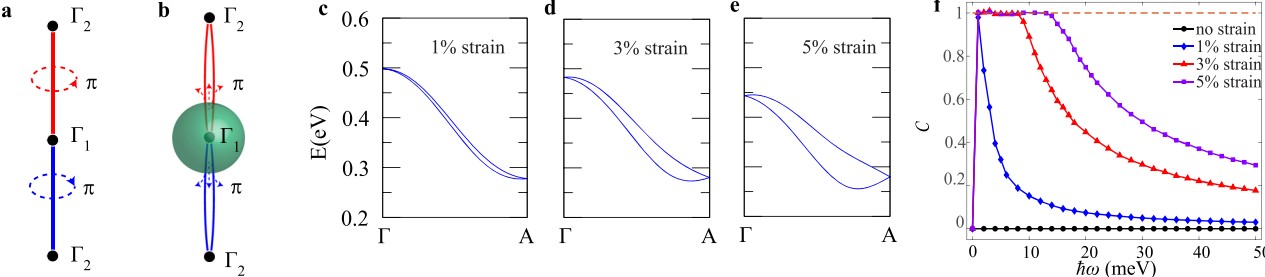

**Fig. 4 Strain-induced Kramers Weyl fermions. a** Schematic plot of a KNL (solid line) carrying Berry flux $\pi$. **b** The Berry flux emerges from TRIMs when the degeneracy of the KNL is lifted. The total flux through a sphere (in green) that enclose the TRIM is $2\pi$. **c–e** The splitting along $\Gamma$–$A$ with 1%, 3%, and 5% strain strengths, respectively. **f** The chiral charge $\mathcal{C}$ versus light frequency $\omega$, calculated at four different strain strengths: no strain (in black), 1% strain (in blue), 3% strain (in red), and 5% strain (in purple).

Therefore, straining achiral crystals provides a new way to create KWSs. In Fig. 4f, we demonstrate how the chiral charge $\mathcal{C}$ of this strain-induced Kramer Weyl point can be measured by the circular photogalvanic effect[64]. It is clear that when a Kramers Weyl point is created, the system shows the quantized circular photogalvanic effect. The details are given in the Supplementary Note 6.

## Discussion

In this work, we point out that all non-centrosymmetric achiral crystals possess KNLs, which connect TRIMs across the whole Brillouin zone. It is important to note that the KNLs are very different from nodal lines generated by band inversions, which can only be accessed in a very small range of energy window[38–41]. As illustrated in the band structure calculation of Fig. 2d, j, KNLs appear in all the bands connecting some TRIMs. These KNLs create the spindle torus-type and the octdong-type Fermi surface as long as the Fermi surfaces enclose TRIMs at an arbitrary Fermi energy. As listed in Table 1, a large number of existing materials are indeed KNLMs. Moreover, generic nodal lines formed by band inversion[87] can be removed without breaking any symmetries. In sharp contrast, the KNLs are enforced and protected by a combination of the time-reversal symmetry and achiral crystal symmetries. The KNLs cannot be removed unless these symmetries are broken.

Here, we briefly discuss some other possible physical consequences of KNLMs, when the KNLs are gapped out. One way to gap out the KNLs is by shining a circularly polarized light on the material, which breaks time-reversal symmetry and in principle can lift the degeneracy of KNLs. This can result in sizable Berry curvature around the KNLs and lead to a light-induced anomalous Hall effect as in the case of graphene[88], where anomalous Hall current arises due to the finite Berry curvature from the light-induced gapped Dirac cone. However, due to the large number of two-dimensional massless Dirac fermions in the material, we expect the effect is larger than that in graphene. Another possibility is to gap out the KNL through a Zeeman field, which can give rise to a field-induced anomalous Hall effect.

So far, we have only discussed KNLs in symmorphic crystals in detail. Indeed, KNLs also appear in all crystals that are non-centrosymmetric and nonsymmorphic. Particularly, there are always KNLs coming out of the $\Gamma$ points of nonsymmorphic crystals. Therefore, we conclude that all non-centrosymmetric achiral crystals possess KNLs, which is the central result of this work. However, the situations in nonsymmorphic crystals are more complicated. For example, as discussed in Supplementary Note 7, screw symmetries can enforce nodal planes at Brillouin boundaries that overwhelm the KNLs in these planes, while glide mirror symmetries can enforce KNLs that are perpendicular to the glide mirror plane at Brillouin zone boundaries. Furthermore, bands at TRIM with higher-fold (such as fourfold and eightfold) degeneracy are widely supported in nonsymmorphic crystals. For example, the TRIM R in nonsymmorphic SG no. 218 ($P\bar{4}3n$) and the TRIM H in nonsymmorphic SG no. 220 ($I\bar{4}3d$) allows eight-dimensional corepresentations, which is consistent with the work of Wieder et al.[23] and Bradlyn et al.[51]. As excepted, in these cases, the KNLs still emerge from these achiral TRIMs as shown in Supplementary Note 7. Specifically, the eightfold degeneracies at the TRIM H in SG. 220 ($I\bar{4}3d$) split into four nondegenerate bands and two KNLs along H–P directions, or four KNLs along H–N and H–$\Gamma$ directions. However, a complete understanding of how the KNLs appear in nonsymmorphic achiral crystals requires more study in the future.

## Methods

**k · p Hamiltonians near TRIMs with achiral little group symmetry**. In this section, we provide the general forms of the **k · p** Hamiltonians near the TRIM points of symmorphic crystals, with achiral little group symmetry to help to understand how KNLs emerge from TRIMs, as listed in Table 2. It is important to note that these **k · p** Hamiltonians can also describe the $\Gamma$ point of nonsymmorphic crystals.

In Table 2, we enumerate all allowed irreducible corepresentations of the ten non-centrosymmetric achiral point groups, the corresponding **k · p** Hamiltonians, as well as directions of KNLs. Here, we use the convention given in ref. [72], where the irreducible representations of AGs are introduced, to label the time-reversal invariant corepresentations. To summarize, we note that (1) there are doubly degenerate KNLs emerging from all TRIM points with achiral little group symmetry. (2) KNLs lie along high symmetry directions in most point groups except certain irreducible corepresentations in $C_{1v}$, $C_{3v}$, and $C_{3h}$, in which cases the KNLs can be pinned along some generic directions within mirror-invariant planes as denoted by the symbol $\in m$. (3) All the irreducible corepresentations are two-dimensional except for the $T_d$ point group that allows a four-dimensional corepresentation. The general form of this four-dimensional Hamiltonian is expressed with $J_i$, which is the angular momentum operators of $J = 3/2$ states, with $i = x, y, z$. It is important to note that there are doubly degenerate KNLs emerging from TRIMs with four-dimensional corepresentations.

Next, we apply Table 2 to understand the KNLs in the band structure of some realistic materials. In BiTeI, the TRIMs $\Gamma$ and A respect $C_{3v}$ symmetry, which allows time-reversal invariant corepresentations $G_{12}^4 : R_3R_4$ and $G_{12}^4 : R_6$. For energy bands at TRIMs described by corepresentations $R_6$, the **k · p** Hamiltonian of

**Table 2 The k · p Hamiltonians at TRIMs with non-centrosymmetric achiral little groups.**

| Point group | IR coreps[72] | $d$ | k · p Hamiltonian | Directions of KNLs |
|---|---|---|---|---|
| $C_{1v}$ | $G_4^1 : R_2R_4$ | 2 | $\alpha_{13}k_z\sigma_x + \alpha_{23}k_z\sigma_y + (\alpha_{31}k_x + \alpha_{32}k_y)\sigma_z$ | $\in m$ |
| $C_{2v}$ | $G_8^5 : R_5$ | 2 | $\alpha_{12}k_y\sigma_x + \alpha_{21}k_x\sigma_y$ | $\hat{z}$ |
| $S_4$ | $G_8^1 : R_2R_8, R_4R_6$ | 2 | $(\alpha_{11}k_x + \alpha_{12}k_y)\sigma_x + (\alpha_{12}k_x - \alpha_{11}k_y)\sigma_y$ | $\hat{z}$ |
| $C_{4v}$ | $G_{16}^{14} : R_6, R_7$ | 2 | $\alpha_{12}k_y\sigma_x - \alpha_{12}k_x\sigma_y$ | $\hat{z}$ |
| $D_{2d}$ | $G_{16}^{14} : R_6, R_7$ | 2 | $\alpha_{11}k_x\sigma_x - \alpha_{11}k_y\sigma_y$ | $\hat{z}$ |
| $C_{3v}$ | $G_{12}^4 : R_3R_4$ | 2 | $i\alpha_1(k_+^3 - k_-^3)\sigma_x + (\alpha_2 k_z^3 + \alpha_4(k_+^3 + k_-^3))\sigma_y + i\alpha_5(k_+^3 - k_-^3)\sigma_z$ | $\in m$ |
| | $G_{12}^4 : R_6$ | 2 | $\alpha_{12}k_y\sigma_x - \alpha_{12}k_x\sigma_y$ | $\hat{z}$ |
| $C_{3h}$ | $G_{12}^1 : R_4R_{10}, R_6R_8$ | 2 | $(\beta_1 k_+^2 + \beta_1^* k_-^2)k_z\sigma_x + i(\beta_1 k_+^2 - \beta_1^* k_-^2)k_z\sigma_y + (\beta_2 k_+^3 + \beta_2^* k_-^3)\sigma_z$ | $\hat{z}, \in m$ |
| | $G_{12}^1 : R_2R_{12}$ | 2 | $(\alpha_1 k_z^3 + \alpha_2 k_+ k_- k_z)\sigma_x + (\alpha_3 k_z^3 + \alpha_4 k_+ k_- k_z)\sigma_y + (\beta_1 k_+^3 + \beta_1^* k_-^3)\sigma_z$ | $\in m$ |
| $C_{6v}$ | $G_{24}^{11} : R_7, R_8$ | 2 | $\alpha_{12}k_y\sigma_x - \alpha_{12}k_x\sigma_y$ | $\hat{z}$ |
| | $G_{24}^{11} : R_9$ | 2 | $i\alpha_1(k_+^3 - k_-^3)\sigma_x + \alpha_2(k_+^3 + k_-^3)\sigma_y$ | $\hat{z}$ |
| $D_{3h}$ | $G_{24}^{11} : R_7, R_8$ | 2 | $(\alpha_1 k_z^3 + \alpha_2 k_+ k_- k_z)\sigma_y + i\alpha_3(k_+^3 - k_-^3)\sigma_z$ | $\hat{x}, C_3\hat{x}, C_3^2\hat{x}, \hat{z}$ |
| | $G_{24}^{11} : R_9$ | 2 | $(\alpha_1 k_z^3 + \alpha_2 k_+ k_- k_z)\sigma_y + i\alpha_3(k_+^3 - k_-^3)\sigma_z$ | $\hat{x}, C_3\hat{x}, C_3^2\hat{x}$ |
| $T_d$ | $G_{48}^{10} : R_4, R_5$ | 2 | $\alpha(k_x(k_y^2 - k_z^2)\sigma_x + k_y(k_z^2 - k_x^2)\sigma_y + k_z(k_x^2 - k_y^2)\sigma_z)$ | $\hat{x}, \hat{y}, \hat{z}, \pm\hat{x} \pm \hat{y} \pm \hat{z}$ |
| | $G_{48}^{10} : R_8$ | 4 | $\beta\sum_i k_i^2 \hat{J}_i^2 + \gamma\sum_{i\neq j}k_i k_j \hat{J}_i\hat{J}_j + \delta\sum_i k_i(\hat{J}_{i+1}\hat{J}_i\hat{J}_{i+1} - \hat{J}_{i+2}\hat{J}_i\hat{J}_{i+2})$ | $\hat{x}, \hat{y}, \hat{z}, \pm\hat{x} \pm \hat{y} \pm \hat{z}$ |

The point group symmetry, the corresponding abstract group (AG) symbols together with time-reversal invariant irreducible corepresentations (IR coreps) are listed. The general form of the Hamiltonians and the direction of the KNLs are listed. In general, the KNLs lie along some high symmetry directions such as the z-direction. For points groups $C_{1v}$, $C_{3v}$, and $C_{3h}$, the KNLs lie within the mirror planes that is denoted as $\in m$. Here, $k_\pm = k_x \pm ik_y$, the Pauli matrices $\sigma_{x,y,z}$ operate on the spinor basis with $J_z = \pm 1/2$ or $J_z = \pm 3/2$, and $\hat{J}_i$ are the angular momentum operators with $J = 3/2$.

the SOC term is

$$H_{so}(\mathbf{k}) = \alpha_{12}(k_y\sigma_x - k_x\sigma_y), \qquad (8)$$

which allows a degenerate line along $\hat{z}$ direction as listed in Table 2, and explains the KNL $\Gamma$–A in Fig. 2c. Similarly, in $Cr_2AgBiO_8$, the TRIMs $\Gamma$ and $Z$ respect $S_4$ symmetry, and the corresponding time-reversal invariant irreducible corepresentations are $G_8^1$: $R_2R_8$ and $R_4R_6$. For energy bands at TRIMs described by these corepresentations, the $\mathbf{k} \cdot \mathbf{p}$ Hamiltonian of SOC term is

$$H_{so}(\mathbf{k}) = (\alpha_{11}k_x + \alpha_{12}k_y)\sigma_x + (\alpha_{12}k_x - \alpha_{11}k_y)\sigma_y, \qquad (9)$$

which vanishes along $\hat{z}$ direction and is consistent with the support of KNL $\Gamma$–$Z$ shown in Fig. 2g.

As shown in Fig. 2d, j, there are KNLs lying within the mirror plane when TRIMs respect $C_{1v}$ symmetry. This property is also manifested by the $\mathbf{k} \cdot \mathbf{p}$ Hamiltonian. The Hamiltonians near such TRIMs have the form

$$H_{so}(\mathbf{k}) = \alpha_{13}k_z\sigma_x + \alpha_{23}k_z\sigma_y + (\alpha_{31}k_x + \alpha_{32}k_y)\sigma_z, \qquad (10)$$

where the mirror operation is $m_z: z \mapsto -z$. Evidently, the $H_{so}(\mathbf{k})$ vanishes along $(-\alpha_{32}, \alpha_{31}, 0)$, which is a direction within the mirror plane.

In the previous sections, we focused on the KNLs which emerge from twofold degenerate points at TRIMs. However, we note that the $T_d$ point group allows a four-dimensional irreducible corepresentations $G_{48}^{10}$: $R_8$. The $\mathbf{k} \cdot \mathbf{p}$ Hamiltonian in basis spanned by states with total angular momentum $J = 3/2$ and azimuthal quantum number $J_z$ (i.e., $\left|3/2, J_z\right\rangle$ with $J_z = \pm 3/2, \pm 1/2$) can be written as[89]

$$H_{so}(\mathbf{k}) = \beta\sum_i k_i^2\hat{J}_i^2 + \gamma\sum_{i\neq j} k_ik_j\hat{J}_i\hat{J}_j + \delta\sum_i k_i(\hat{J}_{i+1}\hat{J}_i\hat{J}_{i+1} - \hat{J}_{i+2}\hat{J}_i\hat{J}_{i+2}), \qquad (11)$$

where $i = x, y, z$ and $i + 1 = y$ if $i = x$, etc. $\hat{J}_i$ are the $4 \times 4$ matrices of the $J = 3/2$ angular momentum operators. This Hamiltonian results in KNLs along $\hat{x}, \hat{y}, \hat{z}$ and $\pm\hat{x}\pm\hat{y}\pm\hat{z}$. This is consistent with the KNLs found in HgSe (SG no. 216, $F\bar{4}3m$) and YPtBi (SG no. 216, $F\bar{4}3m$), as shown in the Supplementary Note 7. It can be seen from the band structure calculations that the four-dimensional corepresentations are decomposed into two two-dimensional irreducible representations along $\Gamma$–X, and one two-dimensional irreducible representation plus two one-dimensional representations along $\Gamma$–L.

## Data availability

The data that support the findings of this study are available from the corresponding author upon reasonable request.

## Code availability

The computer codes that support the findings of this study are available from the corresponding author upon reasonable request.

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

## Acknowledgements

The authors thank the discussions with Zhijun Wang, Quansheng Wu, Andrei Bernevig, Xi Dai, Joel Moore, Titus Neupert, Adrian Po, and Binghai Yan. K.T.L. acknowledges the support of the Croucher Foundation, the Dr. Tai-chin Lo Foundation, and the HKRGC through grants C6025-19G, RFS2021-6S03, 16310219, 16309718, and 16310520.

## Author contributions

K.T.L. conceived the idea of Kramers nodal lines and supervised the project. Y.-M.X. and X.-J.G. performed the major part of the calculations and material analysis. Y.-M.X., X.-J.G., and K.T.L. wrote the manuscript with contributions from all authors. X.Y.X. performed the DFT calculations for BiTeI. C.-P.Z., J.-X.H., and J.Z.G. contributed to part of the calculations and are involved in discussions.

## Competing interests

The authors declare no competing interests.
