## [Peer Review File · Nature Communications]

REVIEWER COMMENTS

Reviewer #1 (Remarks to the Author):

The manuscript by Xie et al. presents a theoretical proposal of the “Kramers Nodal line metals” in achiral noncentrosymmetric materials. This is a very interesting extension of the “Kramers Weyl fermions” predicted in chiral noncentrosymmetric materials proposed earlier. Xie et al found novel properties such as unconventional Fermi surfaces (spindle torus type and the octadong type) and they also predicted quantized optical conductance in the thin film limit. The authors also found a number of interesting materials.

I think that the findings are novel and timely. Their theoretical analysis and calculations are robust and thorough. Therefore, I recommend the paper for publication.

One question is that, the authors may consider to emphasize the distinction between Kramers nodal lines and previous generic nodal lines proposed by others.

Reviewer #2 (Remarks to the Author):

In this work, the authors demonstrate the existence of Kramers-Nodal-Line (KNL) metals, which can be considered the higher-symmetry parent states of the Kramers-Weyl semimetals introduced in [Chang, et al., Nat. Mater. (2018)] (Ref. 55). I think that this work is a scientifically sound and a timely and important addition to the field of symmetry-enforced topological semimetals, and is therefore sufficiently impactful to merit publication in Nature Communications. However, I have some significant reservations with the presentation and terminology employed in the present version of this manuscript, as detailed in my major and minor comments below. If all of my comments were fully addressed, I would likely recommend this work for publication in Nature Communications.

Major Comments:

1. There are some significant issues with the definitions and arguments employed in this work, at least as presented (though I am confident that these issues only arise from the presentation in the current manuscript, and that this work is scientifically sound).

First, it is important to distinguish between crystal momenta k -- like TRIM points -- and the electronic states present in the energy spectrum at k . Throughout this work, the authors use statements like “each TRIM is a Weyl point called Kramers Weyl point [sic].” However, the correct statement is that “each TRIM point hosts electronic states that transform in the irreducible small corepresentations (coreps) of the little group, and, for chiral little groups that are isomorphic to crystallographic point groups modulo translations, states that transform in 2D coreps correspond to Kramers-Weyl fermions (though other coreps also describe chiral fermions, such as the fourfold-degenerate (multifold) fermions in CoSi).”

Second, it is important to emphasize the difference between point groups and little groups. Point groups are finite groups that characterize 0D objects. Little groups are infinite groups that are isomorphic to space groups, and characterize k points in (here) 3D crystals. The point, as emphasized in Refs. 55 and 74, is that in a symmorphic crystal, the little groups can be decomposed into point group operations and integer lattice translations. Hence the small coreps of the little groups in symmorphic crystals are equivalent to the coreps of 0D point groups.

In general, the distinction between, states, crystal momenta, irreps/coreps of point groups, and small irreps/coreps of little groups should be made much more clearer, ideally adopting and adapting the terminology employed in Topological Quantum Chemistry [Bradlyn, et al., Nature (2017)] and Refs. 23, 55, 57-59. When correctly restated, it should be made clear that every noncentrosymmetric crystal with SOC can be a KNL semimetal, but that the specific details depend on which bands transforming in which small coreps are at the Fermi level (as well as the band dispersion/energetics). However, and crucially, the correct statement -- as a corollary to the results of Ref. 55 -- is that every achiral, noncentrosymmetric crystal with relevant SOC is capable of being a KNL semimetal, because every achiral, noncentrosymmetric little group has a 2D corep whose $k \cdot p$ Hamiltonian is that of one point along a KNL.

2. In any space group (SG) and real-material band structure that is left invariant under the symmetries of that space group, it is important to acknowledge that the degeneracy at a given high-symmetry k point can be higher than 2, and depends on the details of the orbitals present at the Fermi energy in the atoms that comprise the crystal. Hence, in Ref. 55, it was emphasized that not every material with a symmorphic chiral SG is a Kramers-Weyl metal -- a significant additional detail was the statement that Kramers-Weyl points, as opposed to chiral multifold fermions -- lie at the Fermi level (however, the statement that all multifold fermions in structurally chiral crystals are topologically chiral was also established in Ref. 55, separate from the definition of Kramers-Weyl points).

In this work, throughout the main text, the authors must discuss the $k \cdot p$ Hamiltonians corresponding to every small corep of every little group in achiral, noncentrosymmetric crystals, establish which coreps (presumably the 2D ones) correspond to KNLs, and, most importantly, explicitly discuss the topology of and introduce terminology for the higher-dimensional (e.g. 4D) coreps in crystals with higher-symmetry little groups.

In order to claim that all noncentrosymmetric, achiral metals with SOC are topological, this step is specifically required, because in some materials, the higher-dimensional coreps, rather than the 2D coreps, lie at the Fermi energy (e.g. HgSe, as discussed in the SM).

3. The presentation of the results in the main text is a bit weaker than in the SM, likely representing one attempt to fit the text within the length limit of an article in Nat. Comm. Many of the arguments in the main text, such as the specialization to symmorphic space groups, appear to be vague, but are eventually supported by stronger statements in the SM that are only obliquely mentioned in the main text. I recommend that the authors move the $k \cdot p$ Hamiltonian discussion in some form and more of the materials analysis to the main text, and try to streamline and strengthen the physical arguments in the main text by reducing the level of detail and more concretely summarizing the details left to the SM.

Minor comments:

1. In 3D (i.e. the bulk of crystalline solids), fourfold point degeneracies are typically referenced as Dirac fermions, and twofold degeneracies are typically labeled Weyl points (if chiral). In the present work, the authors use the term "Dirac" to describe degeneracies -- however, it is unclear in some cases whether the degeneracies are twofold or fourfold. The authors should precisely and clearly establish any usage of the terms "Dirac" and "Weyl" and employ these terms consistently.

2. Due to the Nielsen-Ninomiya theorem, it is impossible to isolate a single chiral fermion at a constant electronic filling at all k points. Therefore, the claim in the abstract that strain drives BiTeI to exhibit an isolated Kramers-Weyl point at the Fermi level is a bit surprising. The authors should make clear in the abstract whether this Kramers-Weyl point is connected to Kramers-Weyl points at higher/lower energies, or other point-like degeneracies, such that readers can understand the compensating distribution of chiral charges (for example, if the Fermi level only encloses one Kramers-Weyl point, then all surface Fermi arcs will be obscured because of Rashba-like band dispersion in all directions).

3. The authors should use the terms "Fermi pocket" and "Fermi surface" more consistently. In general, in the bulk, there is only one Fermi surface, which may consist of disconnected Fermi pockets. I recommend that the authors improve the usage of this terminology upon resubmitting this manuscript.

4. In Sec. II A, the authors begin by stating that Kramers' theorem requires that states at a k point with $T^2 = -1$ are doubly degenerate. However, this is not fully correct -- the precise statement is that with $T^2 = -1$, states must only be at least doubly degenerate, but the degeneracy may be higher depending on the symmetries and coreps in which the degenerate states transform. The authors should more precisely state that they are analyzing only the case of a 2D corep and minimal twofold degeneracy in Sec. II A (as well as throughout the manuscript in any sections that include arguments of the form of those presented in Sec. II A).

5. Throughout this work, the term "reciprocal vector" should be replaced with the term "reciprocal lattice vector."

6. In Sec. II A following Eq. 4, the authors note that the $k \cdot p$ Hamiltonians corresponding to the coreps of achiral, noncentrosymmetric point groups are either "Dirac" fermions or "higher order Dirac points." This terminology is confusing -- if the degeneracies are nondispersing in one direction, then they are not 3D Dirac points, but correspond to one point along a Dirac nodal line. Also, for higher degeneracies, the term "achiral multifold fermion," should be employed instead of "higher-order Dirac point" (presuming that the point degeneracy is indeed dispersing in all 3 directions), as the term "higher-order" with regard to Dirac fermions is used in modern works to characterize 3D Dirac points with higher-order topological 1D hinge states (see [Wieder, et al., Nat. Comm. (2020)], for example).

7. For all instances in which the authors list SGs, the authors should write both the SG number (e.g. 216) as well as the SG symbol (e.g. F-43m).

8. As discussed in Major Comment 1, the X and N points (and all other high-symmetry points) in real materials are not Kramers-Weyl points -- the correct statement is that the Bloch states at a particular energy and k point in the highlighted material correspond to Kramers-Weyl points. All instances in which k points are stated to be point or nodal line fermions should be corrected when this manuscript is revised.

9. The term "proper surfaces" on page 4 is a bit vague, the authors should precisely define the term "proper" (I believe that the authors mean correctly chosen, but in this case, "correctly" is not well defined).

10. In Fig. 3e, it is not clear what protects the twofold degeneracies at $E=0$. The authors should label the states in this figure with mirror (and, if well-defined, rotation) eigenvalues, and explain the pattern of protected degeneracies more clearly.

11. On page 6, the authors claim that KNLs at the Fermi level allow access to "novel physical

properties.” However, this is a bit vague -- the authors should explain how KNLs, versus all other forms of previously established stable nodal lines in solid-state materials, are uniquely capable of hosting the physical properties highlighted by the authors.

12. Berry phase is only meaningfully quantized as a 2π -valued quantity in the absence of chiral symmetry and outside of two-band models (see Ref. 22 and [Ahn, Park, Yang, PRX (2019)], for example). Hence, the only robust statement is likely that a loop around a KNL has either π or 0 Berry phase mod 2π , depending on whether the KNL is twofold degenerate, or is enhanced by crystal symmetry. The authors should revise all statements about integer-valued Berry phases accordingly, paying careful detail to nodal lines that have higher degeneracy at TRIM points (if such forms of KNLs can be symmetry-stabilized).

13. The authors claim on the right-hand side of page 7 that Kramers-Weyl points are difficult to find near the Fermi energy. However, Kramers-Weyl points do lie near the Fermi energy in many materials. For example, in the well-studied CoSi family (Refs. 58 and 59), there are Kramers-Weyl points just below the fourfold multifold fermion at the Fermi level at the Gamma point.

14. On page 8, the response of KNL semimetals is compared to that of graphene. However, graphene is 2D, whereas KNL semimetals are 3D. This comparison needs to be explained in significantly more detail.

15. Also, in the same sentence referenced in Minor Comment 14, the authors claim that KNLs have “a large number of massless Dirac fermions.” It is unclear what is the scale for “large,” and whether the Dirac points are 2D or 3D. In general, one would expect that in idealized KNL semimetals, there are only four 3D KNLs at the Fermi level, and no well-defined 3D Dirac fermions. I recommend that the authors substantially clarify this claim in the revised manuscript.

16. In the text following Eq. S10 in the SM, the authors begin to discuss rotoinversion symmetries using the notation “ S_n .” Naively, one would think that $S_n = I * C_n$. However, the authors seem to be employing a different definition in which $S_3 = I * C_6$. In the revised manuscript, the authors should precisely state the definition of S_n where the symbol first appears in the text.

17. It is unclear whether Table S1 is describing purely magnetic (unitary) irreps, or T-invariant coreps (which may consist of doubled or paired irreps from the magnetic (unitary) subgroup of the little group). In general, all tables should characterize the (small) coreps of T-invariant little groups, and not the irreps of unitary magnetic subgroups of those little groups.

18. Directly preceding Eq. S17, the authors discuss the characters of “relevant” symmetries. In this instance, the word relevant is imprecisely employed -- I believe that the authors should instead use the word “unitary” as a substitute for relevant (or at least a specific, well-defined subset of the unitary symmetries contained within the infinite little group).

19. On pages 8 and 9 of the SM, the authors discuss cases in which BZ-boundary nodal planes stabilized by screw and T symmetries obscure KNLs. However, it is unclear to me if this can also occur in nonsymmorphic SGs with glide symmetries. If the complications related to KNLs only occur a subset of nonsymmorphic SGs - namely those with glide (but not screw) symmetries -- then the authors should analyze the KNL structures that can be stabilized in nonsymmorphic SGs with glide symmetries (or strongly justify why this analysis has not been performed).

20. Throughout the SM, it should briefly be noted that the appearance of Kramers-Weyl points at high-symmetry points in achiral crystals was introduced in Ref. 55.

Response to Reviewers

We thank Referee A for the careful reading of the manuscript and his/her appreciation of the importance of this work. We thank Referee B for a series of questions and comments which helped us to improve the quality of the manuscript.

In the following point-to-point reply, we have addressed all comments from the referees and made revisions in the manuscript and the Supplementary Material accordingly.

Here is the summary of major changes made:

1. We made the definition of some terminology more precise as advised by Referee B.
2. The Discussion Section is expanded to include:
 - a. A paragraph to discuss the distinction between Kramers Nodal lines (KNLs) and generic nodal lines created by band inversion, as advised by Referee A.
 - b. A paragraph to discuss how KNLs can emerge in nonsymmorphic achiral crystals. Explicit examples using the band structures of CsPbF₃ and La₄Bi₃ are given to demonstrate the presence of KNLs in nonsymmorphic achiral crystals.
 - c. A paragraph to discuss how KNLs can emerge in four-fold and eight-fold degenerate time-reversal invariant momentum (TRIM) points (when the states at a TRIM point belong to a higher dimension corepresentation of the little group). A few interesting examples were given in the Supplementary Materials such as HgSe and La₄Bi₃.
3. A Method Section is added to list out the k,p Hamiltonians of the ten noncentrosymmetric achiral point groups. The directions of the KNLs are identified. Importantly, it was shown that KNLs appear even in k,p Hamiltonians belonging to four-dimensional irreducible corepresentations of the point group Td. Explicit examples of KNLs for states belonging to the four-dimensional irreducible corepresentations of Td are given in the Supplementary Material.

The following is the point-by-point reply to the comments of the referees:

Reviewer #1:

[General comments] The manuscript by Xie et al. presents a theoretical proposal of the “Kramers Nodal line metals” in achiral noncentrosymmetric materials. This is a very interesting extension of the “Kramers Weyl fermions” predicted in chiral

noncentrosymmetric materials proposed earlier. Xie et al found novel properties such as unconventional Fermi surfaces (spindle torus type and the octdong type) and they also predicted quantized optical conductance in the thin film limit. The authors also found a number of interesting materials.

I think that the findings are novel and timely. Their theoretical analysis and calculations are robust and thorough. Therefore, I recommend the paper for publication.

[Authors' response]

We sincerely thank the reviewer for the careful reading of the manuscript and his/her strong support for the publication of this work.

[Comments] One question is that, the authors may consider to emphasize the distinction between Kramers nodal lines and previous generic nodal lines proposed by others.

[Authors' response]

We thank the reviewer's suggestion. In the revised manuscript, we expanded the first paragraph in the Discussion Section to emphasize the distinction between Kramers nodal lines (KNLs) and generic nodal lines discovered previously. Specifically, we pointed out that : (i) The generic nodal lines are usually formed by band inversion [Chin. Phys. B 25, 117106 (2016)] and they can be removed without breaking any symmetry. In sharp contrast, the KNLs are enforced and protected by a combination of the time-reversal symmetry and noncentrosymmetric achiral space group symmetries. For example, all states at time-reversal momentum (TRIM) points with achiral point group symmetries in symmorphic noncentrosymmetric achiral crystals, are connected by KNLs. These KNLs cannot be removed unless the time-reversal or the point group symmetries are broken.

(ii) Generic nodal lines usually appear at a local region in the Brillouin zone and a limited energy window. They may or may not have effects on the Fermi surface. KNLs extend across the whole Brillouin zone. For a Fermi surfaces which encloses a TRIM with KNLs, the Fermi surface must cross KNLs, no matter what the Fermi energy is. The KNLs will create nontrivial touching points on the Fermi surface.

Reviewer #2:

Reviewer #2 (Remarks to the Author):

[General comments] In this work, the authors demonstrate the existence of Kramers-Nodal-Line (KNL) metals, which can be considered the higher-symmetry parent states of the Kramers-Weyl semimetals introduced in [Chang, et al., Nat. Mater. (2018)] (Ref. 55). I think that this work is a scientifically sound and a timely and important addition to the field of symmetry-enforced topological semimetals, and is therefore sufficiently impactful

to merit publication in Nature Communications. However, I have some significant reservations with the presentation and terminology employed in the present version of this manuscript, as detailed in my major and minor comments below. If all of my comments were fully addressed, I would likely recommend this work for publication in Nature Communications.

[Authors' response]

We sincerely thank the reviewer for positive comments of our work. The series of comments by the reviewer on the presentation and terminology are very helpful for us to improve the quality of the manuscript.

Major Comments:

1. There are some significant issues with the definitions and arguments employed in this work, at least as presented (though I am confident that these issues only arise from the presentation in the current manuscript, and that this work is scientifically sound).

First, it is important to distinguish between crystal momenta k -- like TRIM points -- and the electronic states present in the energy spectrum at k . Throughout this work, the authors use statements like "each TRIM is a Weyl point called Kramers Weyl point [sic]." However, the correct statement is that "each TRIM point hosts electronic states that transform in the irreducible small corepresentations (coreps) of the little group, and, for chiral little groups that are isomorphic to crystallographic point groups modulo translations, states that transform in 2D coreps correspond to Kramers-Weyl fermions (though other coreps also describe chiral fermions, such as the fourfold-degenerate (multifold) fermions in CoSi).

Second, it is important to emphasize the difference between point groups and little groups. Point groups are finite groups that characterize 0D objects. Little groups are infinite groups that are isomorphic to space groups, and characterize k points in (here) 3D crystals. The point, as emphasized in Refs. 55 and 74, is that in a symmorphic crystal, the little groups can be decomposed into point group operations and integer lattice translations. Hence the small coreps of the little groups in symmorphic crystals are equivalent to the coreps of 0D point groups.

In general, the distinction between, states, crystal momenta, irreps/coreps of point groups, and small irreps/coreps of little groups should be made much clearer, ideally adopting and adapting the terminology employed in Topological Quantum Chemistry [Bradlyn, et al., Nature (2017)] and Refs. 23, 55, 57-59. When correctly restated, it should be made clear that every noncentrosymmetric crystal with SOC can be a KNL semimetal, but that the specific details depend on which bands transforming in which small coreps are at the Fermi level (as well as the band dispersion/energetics). However, and crucially, the correct

statement -- as a corollary to the results of Ref. 55 -- is that every achiral, noncentrosymmetric crystal with relevant SOC is capable of being a KNL semimetal, because every achiral, noncentrosymmetric little group has a 2D corep whose k.p Hamiltonian is that of one point along a KNL.

[Authors' response]

We agree with the reviewer that it is more rigorous to state that each time-reversal invariant momentum (TRIM) that hosts electronic states transforming according to a two-dimensional corepresentations (coreps) of its little group is a Weyl point called Kramers Weyl point. This statement is now made more rigorous in the main text and we thank the referees for pointing it out.

Concerning the difference between point groups and little groups, we thank the reviewer's suggestions. In the revised manuscript, to present the basic concepts more clearly, we have added Section 2A in the beginning of the Supplementary Material. Specifically,

(i) we emphasized the difference between the little groups and the Herring's little group ${}^H G^{\mathbf{k}} = G^{\mathbf{k}}/T^{\mathbf{k}}$ (c.f. Ref. [S1]). In a symmorphic space group case, the Herring's little group ${}^H G^{\mathbf{k}}$ is always isomorphic to a crystalline point group.

“The little group $G^{\mathbf{k}}$ of a wave vector \mathbf{k} is formed by the set of space group operations $\{R_{\alpha}|\mathbf{t}\}$ such that $R_{\alpha}\mathbf{k} = \mathbf{k} + \mathbf{G}_i$, \mathbf{G}_i is the reciprocal lattice vector. For our purpose, it is sufficient to determine the reps of Herring's little group ${}^H G^{\mathbf{k}} = G^{\mathbf{k}}/T^{\mathbf{k}}$ [S1], where $T^{\mathbf{k}}$ is the group of the translational symmetry operations $\{E|\mathbf{t}\}$ with $\exp(-i\mathbf{k} \cdot \mathbf{t}) = 1$. The ${}^H G^{\mathbf{k}}$ in general can be identified with one of the abstract groups (AGs) given in [S1]. For a symmorphic space group, the Herring's little group ${}^H G^{\mathbf{k}}$ is always isomorphic to a point group $G_{\mathbf{k}}$.”

(ii) We emphasized when there is an additional anti-unitary symmetry, such as time-reversal symmetry, the coreps should be considered, and when this anti-unitary symmetry would introduce extra degeneracy.

“When there exhibits an additional anti-unitary symmetry, such as time-reversal symmetry, that leaves \mathbf{k} invariant, the symmetry group becomes $G_{\mathbf{k}} + TG_{\mathbf{k}}$. In this case, the degeneracies are determined by the corepresentations given by the symmetry group $G_{\mathbf{k}} + TG_{\mathbf{k}}$. According to the theory of corepresentations [S1], if a representation D^{Γ_i} is real or complex, the corepresentations is irreducible and the degeneracies at TRIMs will be doubled due to this anti-unitary symmetry, while if D^{Γ_i} is pseudoreal, the corepresentation becomes reducible and there is no extra degeneracy from this anti-unitary symmetry. A more systematic introduction to corepresentations can be found in Ref. [S1].”

[S1] Bradley, C. J. & Cracknell, A. P. *The Mathematical Theory of Symmetry in Solids* (Oxford University Press, 2009).

We also thank the reviewer’s suggestion on the statement of the KNLM. We agree with the reviewer that an achiral noncentrosymmetric crystal with SOC can be a KNL metal, but it happens only when the Fermi surface encloses TRIM points with KNLs. In the revised manuscript, we modified the statement as:

“In this work, we show that all achiral non-centrosymmetric materials with SOC **can be** a new class of topological materials, which we term Kramers nodal line metals (KNLMs). In KNLMs, there are doubly degenerate lines, which we call Kramers nodal lines (KNLs), connecting time-reversal invariant momenta...”

We added the word “can” to highlight the fact that for the material to be a KNLM, the Fermi surface must enclose some time-reversal invariant momenta.

We also fully agree with the referee that there can be bands belonging higher dimensional corepresentations at TRIMs. This is a very important comment. It is important to note that there are always doubly degenerate KNLs coming out of TRIM points with achiral little point groups, *even if the states at the TRIM point are described by a higher dimensional corepresentation*.

More specifically, for noncentrosymmetric symmorphic crystals, all TRIM points have 2-dimensional co-representations except TRIM points with little group symmetry Td which have 4-dimensional co-representations. From k.p analysis (in the newly added Method Section), we explicitly show that there are KNLs coming out of states at TRIM points with Td symmetry. Our claim is further strengthened by DFT calculations. As an illustration, we show that for HgSe, YPtBi, there are doubly degenerate KNLs coming out of the Gamma point (See Fig.R1 below) when the bands are 4-fold degenerate at the Gamma point. Therefore, for noncentrosymmetric symmorphic crystals, regardless of the dimension (2-dimension or 4-dimension) of the co-representations of the little groups, there are always KNLs coming out of the TRIMs with achiral little group symmetry.

Fig. R1. The band structures of HgSe (SG No. 216 $F\bar{4}3m, Td$), half-Heusler material YPtBi (SG No. 216 $F\bar{4}3m, Td$). The KNLs are highlighted in blue color. The four-fold degenerate points at Gamma are highlighted by dashed circles. It is interesting to note that there is one KNL and two KNLs coming out of the four-fold degenerate Gamma point near the Fermi energy along Gamma-L and Gamma-X, respectively.

Nonsymmorphic crystals are more complicated as there are higher dimensional co-representations or nodal planes enforced by nonsymmorphic symmetry operations at Brillouin zone boundaries. However, our analysis is still valid to describe the Gamma point of nonsymmorphic crystals and there are always KNLs coming out of the Gamma point of nonsymmorphic crystals.

Therefore, in general, there are KNLs in all noncentrosymmetric achiral crystals. When the Fermi surface encloses TRIM points connected by KNLs, the KNLs force nontrivial touching points on the Fermi surface (such as in spindle and octadong Fermi surfaces), and the material is a KNL metal.

2. In any space group (SG) and real-material band structure that is left invariant under the symmetries of that space group, it is important to acknowledge that the degeneracy at a given high-symmetry k point can be higher than 2, and depends on the details of the orbitals present at the Fermi energy in the atoms that comprise the crystal. Hence, in Ref. 55, it was emphasized that not every material with a symmorphic chiral SG is a Kramers-Weyl metal -- a significant additional detail was the statement that Kramers-Weyl points, as opposed to chiral multifold fermions -- lie at the Fermi level (however, the statement that all multifold fermions in structurally chiral crystals are topologically chiral was also established in Ref. 55, separate from the definition of Kramers-Weyl points).

In this work, throughout the main text, the authors must discuss the k.p Hamiltonians corresponding to every small corep of every little group in achiral, noncentrosymmetric crystals, establish which coreps (presumably the 2D ones) correspond to KNLs, and, most importantly, explicitly discuss the topology of and introduce terminology for the higher-dimensional (e.g. 4D) coreps in crystals with higher-symmetry little groups.

In order to claim that all noncentrosymmetric, achiral metals with SOC are topological, this step is specifically required, because in some materials, the higher-dimensional coreps, rather than the 2D coreps, lie at the Fermi energy (e.g. HgSe, as discussed in the SM).

[Authors' response]

We thank the reviewer's valuable suggestion.

As explained in the above reply, for noncentrosymmetric symmorphic crystals, there are also KNLs coming out of TRIM points with states belonging to the four-dimensional irreducible corep. The k.p Hamiltonians for states belonging to Td symmetry with four-dimensional irreducible coreps are now added to the Method Section (Table II). In addition to HgSe, we also added the band structure calculations for YPtBi and show that there are KNLs coming out of a four-fold degenerate Gamma point near the Fermi energy (As illustrated in Fig.R1 above).

TABLE II. The $k \cdot p$ Hamiltonians at TRIMs with noncentrosymmetric achiral little groups. The point group symmetry, the corresponding abstract group symbols together with time-reversal invariant irreducible corepresentations are listed. The general form of the Hamiltonians and the direction of the KNLs are listed. In general, the KNLs lie along some high symmetry directions such as the z -direction. For points groups C_{1v} , C_{3v} and C_{3h} , the KNLs lie within the mirror planes which is denoted as $\in m$. Here $k_{\pm} = k_x \pm ik_y$, the Pauli matrices $\sigma_{x,y,z}$ operate on the spinor basis with $J_z = \pm 1/2$ or $J_z = \pm 3/2$, and \hat{J}_i are the angular momentum operators with $J = 3/2$.

Point group	IR coreps [76]	d	$k \cdot p$ Hamiltonian	Directions of KNLs
C_{1v}	$G_4^1 : R_2 R_4$	2	$\alpha_{13} k_z \sigma_x + \alpha_{23} k_z \sigma_y + (\alpha_{31} k_x + \alpha_{32} k_y) \sigma_z$	$\in m$
C_{2v}	$G_8^5 : R_5$	2	$\alpha_{12} k_y \sigma_x + \alpha_{21} k_x \sigma_y$	\hat{z}
S_4	$G_8^1 : R_2 R_8, R_4 R_6$	2	$(\alpha_{11} k_x + \alpha_{12} k_y) \sigma_x + (\alpha_{12} k_x - \alpha_{11} k_y) \sigma_y$	\hat{z}
C_{4v}	$G_{16}^{14} : R_6, R_7$	2	$\alpha_{12} k_y \sigma_x - \alpha_{12} k_x \sigma_y$	\hat{z}
D_{2d}	$G_{16}^{14} : R_6, R_7$	2	$\alpha_{11} k_x \sigma_x - \alpha_{11} k_y \sigma_y$	\hat{z}
C_{3v}	$G_{12}^4 : R_3 R_4$	2	$i\alpha_1 (k_+^3 - k_-^3) \sigma_x + (\alpha_2 k_z^3 + \alpha_4 (k_+^3 + k_-^3)) \sigma_y + i\alpha_5 (k_+^3 - k_-^3) \sigma_z$	$\in m$
	$G_{12}^4 : R_6$	2	$\alpha_{12} k_y \sigma_x - \alpha_{12} k_x \sigma_y$	\hat{z}
C_{3h}	$G_{12}^1 : R_4 R_{10}, R_6 R_8$	2	$(\beta_1 k_+^2 + \beta_1^* k_-^2) k_z \sigma_x + i(\beta_1 k_+^2 - \beta_1^* k_-^2) k_z \sigma_y + (\beta_2 k_+^3 + \beta_2^* k_-^3) \sigma_z$	$\hat{z} \& \in m$
	$G_{12}^1 : R_2 R_{12}$	2	$(\alpha_1 k_z^3 + \alpha_2 k_+ k_- k_z) \sigma_x + (\alpha_3 k_z^3 + \alpha_4 k_+ k_- k_z) \sigma_y + (\beta_1 k_+^3 + \beta_1^* k_-^3) \sigma_z$	$\in m$
C_{6v}	$G_{24}^{11} : R_7, R_8$	2	$\alpha_{12} k_y \sigma_x - \alpha_{12} k_x \sigma_y$	\hat{z}
	$G_{24}^{11} : R_9$	2	$i\alpha_1 (k_+^3 - k_-^3) \sigma_x + \alpha_2 (k_+^3 + k_-^3) \sigma_y$	\hat{z}
D_{3h}	$G_{24}^{11} : R_7, R_8$	2	$(\alpha_1 k_z^3 + \alpha_2 k_+ k_- k_z) \sigma_y + i\alpha_3 (k_+^3 - k_-^3) \sigma_z$	$\hat{x}, C_3 \hat{x}, C_3^2 \hat{x}, \hat{z}$
	$G_{24}^{11} : R_9$	2	$(\alpha_1 k_z^3 + \alpha_2 k_+ k_- k_z) \sigma_y + i\alpha_3 (k_+^3 - k_-^3) \sigma_z$	$\hat{x}, C_3 \hat{x}, C_3^2 \hat{x}$
T_d	$G_{48}^{10} : R_4, R_5$	2	$\alpha(k_x(k_y^2 - k_z^2) \sigma_x + k_y(k_x^2 - k_z^2) \sigma_y + k_z(k_x^2 - k_y^2) \sigma_z)$	$\hat{x}, \hat{y}, \hat{z}, \pm \hat{x} \pm \hat{y} \pm \hat{z}$
	$G_{48}^{10} : R_8$	4	$\beta \sum_i k_i^2 \hat{J}_i^2 + \gamma \sum_{i \neq j} k_i k_j \hat{J}_i \hat{J}_j + \delta \sum_i k_i (\hat{J}_{i+1} \hat{J}_i \hat{J}_{i+1} - \hat{J}_{i+2} \hat{J}_i \hat{J}_{i+2})$	$\hat{x}, \hat{y}, \hat{z}, \pm \hat{x} \pm \hat{y} \pm \hat{z}$

For nonsymmorphic noncentrosymmetric achiral crystals, the k.p Hamiltonian description is still applicable for the Gamma points. Therefore, there are always KNLs coming out of the Gamma points (which always have 2-dimensional or 4-dimensional coreps). KNLs coming out of the Gamma points for nonsymmorphic crystals CsPbF₃ and La₄Bi₃ are shown in the Fig.R2 (c) and (d) below, which are expected for a TRIM with C_{3v} and Td symmetry, respectively.

Therefore, it is generally valid to state that there are KNLs in all achiral crystals.

For nonsymmorphic achiral crystals, there can be TRIM points with 4-dimensional and even 8-dimensional coreps at the Brillouin zone boundaries as listed in Table S5 of the Supplementary Materials (copied below). However, in some cases, there can be nodal planes on the Brillouin zone boundaries and KNLs are not well defined on the nodal planes.

But these nodal planes do not affect the conclusion that there are KNLs coming out of the Gamma points in all nonsymmorphic noncentrosymmetric achiral crystals.

Here, we discuss the TRIMs with higher-dimensional coreps in noncentrosymmetric achiral crystals in more details. In general, we can identify the TRIMs with higher-dimensional coreps in noncentrosymmetric achiral crystals from their little group symmetry. We explicitly enumerated the possible higher-dimensional coreps, which are labelled with the irreducible abstract groups, allowed by the Herring's little group ${}^H G^{\mathbf{k}}$ at TRIMs (c.f. [S1]) and summarized symmetry-allowed higher-dimensional coreps at TRIMs for non-magnetic non-centrosymmetric achiral crystals in Table S5 (see below). In symmorphic groups, there allows 4D coreps in TRIMs respecting T_d symmetry, including TRIMs Γ , R in SG No. 215, Γ in SG No. 216 and Γ , H in SG No. 217. In contrast, for nonsymmorphic achiral SGs, because the presence of nonsymmorphic operations (glide mirrors or screw rotations) complicates the algebra, 4D coreps are more widely supported at TRIMs. And notably, the TRIM R in SG No. 218 and the TRIM H in SG No. 220 further allows 8D coreps, which is consistent with the findings of Bradlyn etc. in [Science 353, 5037 (2016)].

TABLE S5. The symmetry-allowed higher-dimensional corepresentations at TRIMs for non-magnetic non-centrosymmetric achiral crystals.

Symmorphic SGs							
SG No.	TRIMs	AGs and coreps under \mathcal{T}	d	SG No.	TRIMs	AGs and coreps under \mathcal{T}	d
215 $F\bar{4}3m$ (T_d)	Γ , R	$G_{48}^{10} : R_8$	4	216 $F\bar{4}3m$ (T_d)	Γ	$G_{48}^{10} : R_8$	4
217 $I\bar{4}3m$ (T_d)	Γ , H	$G_{48}^{10} : R_8$	4				
Nonsymmorphic SGs							
SG No.	TRIMs	AGs and coreps under \mathcal{T}	d	SG No.	TRIMs	AGs and coreps under \mathcal{T}	d
26 $Pmc2_1$, 27 $Pcc2$ (C_{2v})	Z, U, T	$G_{16}^8 : R_9 R_9$	4	29 $Pca2_1$ (C_{2v})	Z, U	$G_{16}^8 : R_9 R_9$	4
30 $Pnc2$, 31 $Pmn2_1$ (C_{2v})	Z, U	$G_{16}^8 : R_9 R_9$	4	32 $Pba2$ (C_{2v})	S, R	$G_{16}^8 : R_9 R_9$	4
33 $Pna2_1$ (C_{2v})	Z, S, R	$G_{16}^8 : R_9 R_9$	4	34 $Pnn2$ (C_{2v})	Z, S	$G_{16}^8 : R_9 R_9$	4
36 $Cmc2_1$ (C_{2v})	Z, T	$G_{16}^8 : R_9 R_9$	4	37 $Ccc2$ (C_{2v})	Z, T	$G_{16}^8 : R_9 R_9$	4
43 $Fmm2$ (C_{2v})	Z	$G_{16}^8 : R_9 R_9$	4	101 $P4_2cm$ (C_{4v})	R	$G_{16}^8 : R_9 R_9$	4
103 $P4cc$ (C_{4v})	Z, A	$G_{32}^{10} : R_6 R_6, R_7 R_7$	4	103 $P4cc$ (C_{4v})	R	$G_{16}^8 : R_9 R_9$	4
104 $P4nc$ (C_{4v})	Z	$G_{32}^{10} : R_6 R_6, R_7 R_7$	4	106 $P4_2bc$ (C_{4v})	A	$G_{32}^{10} : R_6 R_6, R_7 R_7$	4
114 $P\bar{4}2_1c$ (D_{2d})	A	$G_{32}^{10} : R_6 R_6, R_7 R_7$	4	116 $P\bar{4}c2$ (D_{2d})	R	$G_{16}^8 : R_9 R_9$	4
159 $P31c$ (C_{3v})	A	$G_{12}^4 : R_5 R_5$	4	161 $R3c$ (C_{3v})	Z	$G_{12}^4 : R_5 R_5$	4
184 $P6cc$ (C_{6v})	L	$G_{16}^8 : R_9 R_9$	4	184 $P6cc$ (C_{6v})	A	$G_{48}^{12} : R_7 R_7, R_8 R_8, R_9 R_9$	4
185 $P6_3cm$, 186 $P6_3mc$ (C_{6v})	L; A	$G_{16}^8 : R_9 R_9; G_{48}^{13} : R_{14} R_{15}$	4	188 $P\bar{6}c2$ (D_{3h})	A	$G_{48}^{14} : R_{11} R_{12}$	4
190 $P\bar{6}2c$ (D_{3h})	A	$G_{48}^{14} : R_{11} R_{12}$	4	218 $P\bar{4}3n$ (T_d)	Γ ; X	$G_{48}^{10} : R_8; G_{32}^{11} : R_6 R_7$	4
218 $P\bar{4}3n$ (T_d)	R	$G_{96}^7 : R_6 R_7$	4	219 $F\bar{4}3c$ (T_d)	Γ	$G_{48}^{10} : R_8$	4
220 $I\bar{4}3d$ (T_d)	Γ	$G_{48}^{10} : R_8$	4	220 $I\bar{4}3d$ (T_d)	H	$G_{96}^7 : R_6 R_7$	4
218 $P\bar{4}3n$ (T_d)	R	$G_{96}^7 : R_{15} R_{15}$	8	220 $I\bar{4}3d$ (T_d)	H	$G_{96}^7 : R_{15} R_{15}$	8

Next, we present some realistic material examples to verify the results in Table S5 and show how KNLs emerge out from these TRIMs when higher-dimensional coreps are hosted near Fermi level. In Fig. R2, we plotted the band structure of (a) HgSe (SG No. 216 $F\bar{4}3m$, T_d), (b) half-Heusler material YPtBi (SG No. 216 $F\bar{4}3m$, T_d), (c) CsPbF₃ (SG No. 161 $R3c$, C_{3v}), (d) La₄Bi₃ (SG No. 220 $I\bar{4}3d$, T_d). The states at TRIMs described by 4D coreps are circled in (a) to (c) and states described by 8D coreps are circled in (d). It can

be seen that the appearance of higher dimension coreps for these space groups are consistent with Table S5: The TRIM Γ for symmorphic SG No. 216, the TRIM Z for nonsymmorphic SG No. 161 can host 4D coreps, while the TRIM H for nonsymmorphic SG No. 220 can host 8D coreps. The doubly degenerate KNLs in Fig. R2 (a) to (d) are highlighted in blue colour. It is evident that there are KNLs emerging from states at TRIM points with higher dimensional co-representations.

Fig. R2 Example materials that exhibit higher coreps near Fermi level. (a) to (d), respectively, show the band structures of HgSe (SG No. 216 $F\bar{4}3m, Td$), half-Heusler material YPtBi (SG No. 216 $F\bar{4}3m, Td$), CsPbF₃ (SG No. 161 $R3c, C3v$), La₄Bi₃ (SG No. 220 $I\bar{4}3d, Td$). The states at TRIMs described by 4D coreps are circled in (a) to (c) and states described by 8D coreps are circled in (d). The blue curves are doubly degenerate KNLs.

According to the reviewer's suggestions, we have added the discussions of the KNLs from the point view of the corresponding coreps at TRIMs and the corresponding $k \cdot p$ Hamiltonians in the Method Section. Moreover, we have also added the above discussions on higher-dimensional coreps in the Method Section and Section 7F in the Supplementary Material.

3. The presentation of the results in the main text is a bit weaker than in the SM, likely representing one attempt to fit the text within the length limit of an article in Nat. Comm. Many of the arguments in the main text, such as the specialization to symmorphic space groups, appear to be vague, but are eventually supported by stronger statements in the SM that are only obliquely mentioned in the main text. I recommend that the authors move the k.p Hamiltonian discussion in some form and more of the materials analysis to the main text, and try to streamline and strengthen the physical arguments in the main text by reducing the level of detail and more concretely summarizing the details left to the SM.

[Authors' response]

We thank the reviewer's suggestion. We now added the k.p Hamiltonians belonging to all the achiral point groups to the Method Section. In particular, we showed how the KNLs we discussed in the material analysis of the main text can be understood from the k.p Hamiltonians. We also briefly summarized some results for the nonsymmorphic crystals in the discussion part.

Minor comments:

1. In 3D (i.e. the bulk of crystalline solids), fourfold point degeneracies are typically referenced as Dirac fermions, and twofold degeneracies are typically labelled Weyl points (if chiral). In the present work, the authors use the term "Dirac" to describe degeneracies - - however, it is unclear in some cases whether the degeneracies are twofold or fourfold. The authors should precisely and clearly established any usage of the terms "Dirac" and "Weyl" and employ these terms consistently.

[Authors' response]

For a plane in the 3D Brillouin zone which intercepts a KNL, we can obtain a k.p Hamiltonian describing the states near the KNL on the momentum plane. For example, assuming that there exists a KNL along kz direction, for a fixed kz, the k.p Hamiltonian to the lowest order near the KNLs can be written as

$$H(\mathbf{p}) = f_0(\mathbf{p})\sigma_0 + v_1\mathbf{p}_+^m\sigma_+ + v_2\mathbf{p}_-^m\sigma_-, \quad (\text{R1})$$

where \mathbf{p} denotes the momentum perpendicular to the KNL, $\mathbf{p}_\pm = p_x + ip_y$, $\sigma_\pm = \sigma_x \pm \sigma_y$ and $m = 1,2,3$, corresponding to linear-, quadratic-, cubic-band touchings respectively. The term "Dirac" refers to the " $v_1\mathbf{p}_+^m\sigma_+ + v_2\mathbf{p}_-^m\sigma_-$ " part of the Hamiltonian. Note that the Dirac point here is the interception point between the momentum plane and the KNL and it is *always two-fold degenerate*.

For $m=1$, the Hamiltonian of the form " $v_1\mathbf{p}_+^m\sigma_+ + v_2\mathbf{p}_-^m\sigma_-$ " is referred to as a Dirac Hamiltonian.

For $m=2$ or $m=3$, the Hamiltonian of the form $v_1\mathbf{p}_+^m\sigma_+ + v_2\mathbf{p}_-^m\sigma_-$ is referred to as a higher-order Dirac Hamiltonian following the naming of Ref.[Nat. Commun. 5, 4898 (2014)]. We understand that this way of naming has yet to be widely accepted as the study of this kind of Hamiltonians in solid state materials is a relatively new topic. The meaning of Dirac point here is different from the Dirac point which describes a fourfold degenerate point in 3D Dirac materials. To explain the usage of “Dirac” clearly, we added a footnote [71] in the revised manuscript.

2. Due to the Nielsen-Ninomiya theorem, it is impossible to isolate a single chiral fermion at a constant electronic filling at all \mathbf{k} points. Therefore, the claim in the abstract that strain drives BiTeI to exhibit an isolated Kramers-Weyl point at the Fermi level is a bit surprising. The authors should make clear in the abstract whether this Kramers-Weyl point is connected to Kramers-Weyl points at higher/lower energies, or other point-like degeneracies, such that readers can understand the compensating distribution of chiral charges (for example, if the Fermi level only encloses one Kramers-Weyl point, then all surface Fermi arcs will be obscured because of Rashba-like band dispersion in all directions).

[Authors’ response]

Fig. R3 (a) A single Fermi surface encloses one Weyl point below Fermi energy. (b) Two Fermi surfaces with opposition Chern number enclosing one Weyl point below Fermi energy. (c) Two Fermi pockets with opposite Chern number enclose two Weyl points separately. (d) A schematic plot the energy dispersion far from A point in BiTeI near the Fermi energy.

We agree with reviewer that due to the Nielsen-Ninomiya theorem, a single chiral fermion at a constant electronic filling at all \mathbf{k} points, as show in case (a) of Fig. R3, is impossible,

and the typical scenario is that there are two chiral fermions below the Fermi energy (Fig. R3 (b)) or two Fermi surfaces with opposite chiral charges enclose the same Weyl point (Fig. R3 (c)). For the BiTeI we studied, the energy bands only behave as chiral fermions near A point instead of all k points and far from the A point, the energy bands finally bend up and have Rashba-like dispersion (Fig. R3 (d)). Due to this Rashba-like dispersion, there are two Fermi pockets enclosing the A point in strained BiTeI and the two Fermi pockets carry opposite chiral charges. Hence, the Nielsen-Ninomiya theorem is not violated in this case. Since the two Fermi pockets carrying opposite chiral charges enclose the same Kramers Weyl point, the Fermi arcs is indeed obscured as discussed in [Nature Materials 17, 978–985 (2018)].

In the revised manuscript, we added “**Although there is only a single Weyl point near Fermi energy, the Nielsen-Ninomiya theorem is not violated because there are two Fermi pockets carrying opposite chiral charges which enclose this Weyl point.**” in Section D of main text.

3. The authors should use the terms “Fermi pocket” and “Fermi surface” more consistently. In general, in the bulk, there is only one Fermi surface, which may consist of disconnected Fermi pockets. I recommend that the authors improve the usage of this terminology upon resubmitting this manuscript.

[Authors’ response]

In the revised manuscript, we have made the usage of “Fermi pocket” and “Fermi surface” consistently. Specifically, we used the “Fermi surface” to denote the whole Fermi surface, such as the spindle Fermi surface, octadong Fermi surface, while we used “Fermi pocket” for isolated parts of a whole Fermi surface, such as “electron Fermi pockets”, “hole Fermi pockets”.

4. In Sec. II A, the authors begin by stating that Kramers’ theorem requires that states at a k point with $T^2=-1$ are doubly degenerate. However, this is not fully correct -- the precise statement is that with $T^2=-1$, states must only be at least doubly degenerate, but the degeneracy may be higher depending on the symmetries and coreps in which the degenerate states transform. The authors should more precisely state that they are analyzing only the case of a 2D corep and minimal twofold degeneracy in Sec. II A (as well as throughout the manuscript in any sections that include arguments of the form of those presented in Sec. II A).

[Authors’ response]

We agree with the reviewer that the $T^2=-1$ only guarantee that the electronic band is at least doubly degenerate at the TRIMs. In the revised manuscript, we changed this sentences in Sec. II A as

“According to Kramers theorem, each electronic band is **at least** doubly degenerate at a TRIM \mathbf{k}_0 , where $\mathbf{k}_0 = -\mathbf{k}_0 + \mathbf{G}_i$, and \mathbf{G}_i denotes a reciprocal lattice vector. **We first focus on the cases that the energy bands are two-fold degenerate at TRIM points, and the cases with four-fold degeneracy are discussed in the Method Section.**”

And we explicitly specified the dimension of coreps when we discussed KNLs in the later parts.

5. Throughout this work, the term “reciprocal vector” should be replaced with the term “reciprocal lattice vector.”

[Authors’ response]

We thank referee for the suggestion. In the revised manuscript, we replaced the term “reciprocal vector” as “reciprocal lattice vector”.

6. In Sec. II A following Eq. 4, the authors note that the k.p Hamiltonians corresponding to the coreps of achiral, noncentrosymmetric point groups are either “Dirac” fermions or “higher order Dirac points.” This terminology is confusing -- if the degeneracies are nondispersing in one direction, then they are not 3D dirac points, but correspond to one point along a Dirac nodal line. Also, for higher degeneracies, the term “achiral multifold fermion,” should be employed instead of “higher-order Dirac point” (presuming that the point degeneracy is indeed dispersing in all 3 directions), as the term “higher-order” with regard to Dirac fermions is used in modern works to characterize 3D Dirac points with higher-order topological 1D hinge states (see [Wieder, et al., Nat. Comm. (2020)], for example).

[Authors’ response] We thank the reviewer for helping us to have a clearer terminology. As we mentioned in the reply of minor comment 1, for a plane in the 3D Brillouin zone which intercepts a KNL, we can obtain a k.p Hamiltonian which describes the states near the interception of the KNL and the momentum plane. These Hamiltonian have terms of the form “ $v_1\mathbf{p}_+^m\sigma_+ + v_2\mathbf{p}_-^m\sigma_-$ ”, where the $\mathbf{p}_+ = \mathbf{p}_- = 0$ is the Dirac point. For $m=1$, the Hamiltonian of the form “ $v_1\mathbf{p}_+^m\sigma_+ + v_2\mathbf{p}_-^m\sigma_-$ ” is referred to as a Dirac Hamiltonian. For $m=2$ or $m=3$, the Hamiltonian of the form “ $v_1\mathbf{p}_+^m\sigma_+ + v_2\mathbf{p}_-^m\sigma_-$ ” is referred to as a higher-order Dirac Hamiltonian. We want to clarify that the term “higher-order” is related to the energy dispersion near the KNL which is not related to the degeneracies. The degeneracies of the Dirac point is always two-fold as the KNL is two-fold degenerate. We admit that this terminology “higher-Dirac” is not well-accepted and may cause confusion. Hence in the revised manuscript, we add a footnote [71] to explicitly clarify this terminology.

7. For all instances in which the authors list SGs, the authors should write both the SG number (e.g. 216) as well as the SG symbol (e.g. F-43m).

[Authors' response]

We have added the SG symbol accordingly in the revised manuscript.

8. As discussed in Major Comment 1, the X and N points (and all other high-symmetry points) in real materials are not Kramers-Weyl points -- the correct statement is that the Bloch states at a particular energy and k point in the highlighted material correspond to Kramers-Weyl points. All instances in which k points are stated to be point or nodal line fermions should be corrected when this manuscript is revised.

[Authors' response]

In the revised manuscript, we changed it as “the Bloch states for each band near X and N points in Cr₂AgBiO₈ describe Kramers Weyl fermions”. And in the revision, we changed all the statements that directly relate the crystal momentum to the states accordingly.

9. The term “proper surfaces” on page 4 is a bit vague, the authors should precisely define the term “proper” (I believe that the authors mean correctly chosen, but in this case, “correctly” is not well defined).

[Authors' response]

The surface which possesses Fermi arc states is the (001) surface. This is now mentioned in the main text instead of using the word “proper surfaces”.

10. In Fig. 3e, it is not clear what protects the twofold degeneracies at E=0. The authors should label the states in this figure with mirror (and, if well-defined, rotation) eigenvalues, and explain the pattern of protected degeneracies more clearly.

[Authors' response]

We want to clarify that the two bands in Fig. 2e of the main text (see the figure in below) corresponds to $kz=0$ and $kz=\pi$ respectively. Therefore, the crossings at E=0 do not indicate degeneracy.

Fig. R4 (copy of Fig. 2e of the main text) The energy dispersion as a function of k_y for $k_x=0$ and fixed $k_z=0$ (purple) and $k_z=\pi$ (red) respectively.

11. On page 6, the authors claim that KNLs at the Fermi level allow access to “novel physical properties.” However, this is a bit vague -- the authors should explain how KNLs, versus all other forms of previously established stable nodal lines in solid-state materials, are uniquely capable of hosting the physical properties highlighted by the authors.

[Authors’ response]

It is important to note that the KNLs have a very unique property that they connect TRIM points in the Brillouin zone. Therefore, they go across the whole Brillouin zone. When a Fermi surface enclose a TRIM point, the KNLs always create band touching points on the Fermi surface. This results in two types of Fermi surfaces, the spindle and the octdng Fermi surfaces respectively. The octdng Fermi surfaces are particularly interesting, as all the electrons on the octdng Fermi surface can be described by a two-dimensional massless Dirac Hamiltonian when one of the momenta is fixed to be a constant (see Fig. R5 (a) below). We showed that this resulted in novel optical conductivity properties as shown in Fig. 3 of the main text which is also copied below.

Fig. R5 (a) The octdng Fermi surface. On a fixed momentum plane (highlighted red rectangle), the energy dispersion, being Dirac-like, is schematically shown on the right panel. μ indicates the position of the Fermi energy. (b) The bulk optical conductivity for a model material with octdng Fermi surfaces. (c) Schematic plot of optical excitations that contribute to the optical conductivity for the hole-type (electron-type) Dirac fermions with onset frequency ω_1 (ω_2). (d) The step-wise quantized optical conductivity in thin films.

It is important that the appearance of the linearly bulk optical conductivity and the stepwise quantized optical conductivity in thin films depends on the fact that all the electrons on the octdng Fermi surface can be described by massless Dirac Hamiltonians. **However, the electrons on a Fermi surface which encloses a generic nodal line do not have to be described by massless Dirac Hamiltonians.** Therefore, the linearly optical conductivity in the bulk **together with** quantized optical conductivity in the thin film limit do not appear in generic nodal line materials in general [PRB 96, 155150 (2017)].

In the revised manuscript, we emphasize the distinction between Kramers nodal lines and generic nodal lines more in the first paragraph of the discussion part.

12. Berry phase is only meaningfully quantized as a $Z2$ -valued quantity in the absence of chiral symmetry and outside of two-band models (see Ref. 22 and [Ahn, Park, Yang, PRX

(2019)], for example). Hence, the only robust statement is likely that a loop around a KNL has either π or 0 Berry phase mod 2π , depending on whether the KNL is twofold degenerate, or is enhanced by crystal symmetry. The authors should revise all statements about integer-valued Berry phases accordingly, paying careful detail to nodal lines that have higher degeneracy at TRIM points (if such forms of KNLs can be symmetry-stabilized).

[Authors' response]

We thank the reviewer for noting this. We agree with the reviewer that the Berry phase is a Z_2 -valued quantity, and a 2π Berry phase can be gauged out.

In the revised manuscript, we have revised these statements accordingly. For example, we changed the description of Berry phase carries by the KNL as

“When a Bloch electron moves around a KNL adiabatically, it acquires a quantized Berry phase of $m\pi \bmod 2\pi$, and one can regard a KNL carrying Berry curvature flux of π as a Dirac solenoid”.

And we want to clarify the KNL we discussed is always two-fold degenerate and we do not find any KNL that exhibits higher fold degeneracies up to now. The Berry phase is 0 or π of a loop around a KNL, depending on the type of touching. Specifically, the Berry phase is π for KNLs giving by linear and cubic touching, while the Berry phase is zero for KNLs giving by quadratic touching. Hence the value of the Berry phase mainly depends on the touching type of the KNL rather than the degeneracies of TRIMs where the KNL emerges from. Also, we want to emphasize that we used “higher-order Dirac” to indicate the energy dispersion near a KNL rather than to refer to the degeneracies of a KNL.

13. The authors claim on the right-hand side of page 7 that Kramers-Weyl points are difficult to find near the Fermi energy. However, Kramers-Weyl points do lie near the Fermi energy in many materials. For example, in the well-studied CoSi family (Refs. 58 and 59), there are Kramers-Weyl points just below the fourfold multifold fermion at the Fermi level at the Gamma point.

[Authors' response]

We thank the reviewer for pointing this out. In previous version, we stated that “This is in sharp contrast to the Kramers Weyl points discovered in Ref. [55], which are usually far away from the Fermi energy.” To avoid being misleading, we have deleted this statement in the revised manuscript.

14. On page 8, the response of KNL semimetals is compared to that of graphene. However, graphene is 2D, whereas KNL semimetals are 3D. This comparison needs to be explained in significantly more detail.

[Authors' response]

In case of graphene, the Berry curvature appears in two valleys by gapping out the Dirac points through applying circularly polarized light. This can result in ultrafast anomalous Hall currents in experiment [Nat. Phys. 16, 38 (2020)].

Fig. R6 (a) The spindle Fermi surface. The momentum planes that cut through the Fermi surfaces are highlighted and the dispersion on each momentum plane are schematically plotted on the right panel. The states near the crossing point (see the red circle on the right panel) which lies on the KNL can be viewed as a massless Dirac . (b) The octadong Fermi surface. In each momentum plane that cuts through the octadong Fermi surfaces, the electrons can be viewed as Dirac electrons, as schematically shown in the right panel.

In our case, we believe that a similar light induced anomalous Hall effect has a chance to be observed in a KNL metal. The KNL could create the spindle Fermi surface and octadong Fermi surface. As we discussed and shown in Fig. R6, by fixing a momentum, it can be seen that the states near the KNL can be effectively described as a 2D Dirac massless Hamiltonian. The circularly polarized light breaks time-reversal symmetry and can lift the degeneracy of KNLs. And those states originally near the KNL are described by 2D massive Dirac Hamiltonians, which result in sizable Berry curvature and give rise to the anomalous Hall currents. Importantly, from this point of view, a large number of massive Dirac electrons (scale with the system size) are generated, and thus we believe that this light induced anomalous hall effect could actually be more prominent than the graphene studied in [Nat. Phys. 16, 38 (2020)].

In the revised manuscript, we explained this idea in more detail in the discussion part as following:

“One way to gap out the KNLs is by shining a circularly polarized light on the material, which breaks time reversal symmetry and in principle can lift the degeneracy of KNLs. This can result in sizable Berry curvature around the KNLs and lead to a light-induced

anomalous Hall effect as in the case of graphene [91], where anomalous Hall current arises due to the finite Berry curvature from the light-induced gapped Dirac cone.”

15. Also, in the same sentence referenced in Minor Comment 14, the authors claim that KNLs have “a large number of massless Dirac fermions.” It is unclear what is the scale for “large,” and whether the Dirac points are 2D or 3D. In general, one would expect that in idealized KNL semimetals, there are only four 3D KNLs at the Fermi level, and no well-defined 3D Dirac fermions. I recommend that the authors substantially clarify this claim in the revised manuscript.

[Authors’ response]

As we discussed in Section C, here the massless Dirac fermions are 2D, which is defined within a momentum plane which intercepts a KNL. And in this sense, the number of such 2D massless Dirac fermions depends on how many momentum planes we can choose, which is expected to scale with the system size. Hence, we used the term “large”. In the revised manuscript, we added “**The number of two-dimensional massless Dirac fermions are expected to scale with the system size.**” in Section C where we first use the term “large”. And we have changed the “massless Dirac fermions” in main text as “**two-dimensional massless Dirac fermions**” to clarify this.

16. In the text following Eq. S10 in the SM, the authors begin to discuss rotoinversion symmetries using the notation “ S_n .” Naively, one would think that $S_n = I * C_n$. However, the authors seem to be employing a different definition in which $S_3 = I * C_6$. In the revised manuscript, the authors should precisely state the definition of S_n where the symbol first appears in the text.

[Authors’ response]

Yes, we used the Schoenflies notation where S_n represents the combination of a mirror and a n -fold rotation perpendicular to the mirror plane. In the revised manuscript, we explicitly wrote down the definition of S_n in Eq. S11.

17. It is unclear whether Table S1 is describing purely magnetic (unitary) irreps, or T-invariant coreps (which may consist of doubled or paired irreps from the magnetic (unitary) subgroup of the little group). In general, all tables should characterize the (small) coreps of T-invariant little groups, and not the irreps of unitary magnetic subgroups of those little groups.

[Authors' response]

Yes, we considered the coreps of T-invariant little groups since we are considering the non-magnetic crystals. In the revised manuscript, we explicitly mentioned this in the table II of the Method Section, as well as in the SM.

18. Directly preceding Eq S17, the authors discuss the characters of “relevant” symmetries. In this instance, the word relevant is imprecisely employed -- I believe that the authors should instead use the word “unitary” as a substitute for relevant (or at least a specific, well-defined subset of the unitary symmetries contained within the infinite little group).

[Authors' response]

We changed it to “each unitary symmetry operation R ” to make it consistent with the presentation in the main text.

19. On pages 8 and 9 of the SM, the authors discuss cases in which BZ-boundary nodal planes stabilized by screw and T symmetries obscure KNLs. However, it is unclear to me if this can also occur in nonsymmorphic SGs with glide symmetries. If the complications related to KNLs only occur a subset of nonsymmorphic SGs - namely those with glide (but not screw) symmetries -- then the authors should analyze the KNL structures that can be stabilized in nonsymmorphic SGs with glide symmetries (or strongly justify why this analysis has not been performed).

[Authors' response] We thank the reviewer's questions.

Yes, previously, we mentioned nodal degeneracies at the Brillouin zone-boundary can be supported if some screw symmetries are included in the SG symmetry G (c.f. Nature Materials 17, 978–985 (2018) for a more detail discussion on this). Although the glide symmetry is not related to the nodal plane degeneracies at Brillouin zone boundaries, we realized that a glide mirror symmetry will enforce a degenerate line that is perpendicular to the glide mirror plane at the Brillouin zone boundary for non-magnetic crystals (see a detail proof in revised Supplementary Section 7 B).

In the Fig. R7, we present some material examples in which the KNLs enforced by the glide mirror symmetry at the Brillouin zone boundary. In Fig. R7 (c), we show the band structures of TePb_2O_5 (SG No. 9, $C1v$), which possesses only one glide mirror symmetry. The TRIMs that possess the glide mirror symmetry at the Brillouin zone boundary are highlighted by red colour. We display the KNLs within the glide mirror plane in Fig. R7 (d), where we select two SOC-split bands and plot the energy difference with respect to the momentum k as we did in Fig. 2d and 2j of main text. As expected, there is a KNL that emerges from Γ point and connects with another TRIM Y. However, it can be noted that there is no KNL connecting Z and L on this plane, which is sharply contrast with BiPd_2Pb

(SG. No.8 C1v), a material with C1v symmetry but symmorphic space group, shown in main text (see Fig. 2j of main text).

This difference originates from Z and L are the TRIMs that locate on the Brillouin zone boundary and possess glide symmetry. In this case, as we discussed in the revised Supplementary Section 7 B, the KNL emerging from these TRIMs are perpendicular to the glide mirror plane due to the combination of time-reversal and glide mirror symmetry. Indeed, by plotting the energy difference of the selected two bands with respect to the momentum k at the Brillouin zone boundary, we found a KNL ZL_1 that is perpendicular to the glide mirror plane (Fig. R7 (e)). We also checked another the material CsPbF₃ (SG No. 161 R3c, C3v), which we previously presented as an example of nonsymmorphic crystals with KNLs. For this material, there are TRIMs L, Z that possess glide mirror symmetries on the Brillouin zone boundary (see the band structure in Fig. R7 (h)). It can be found the KNL connecting TRIMs Z and L via point B=B₁, which are highlighted in Fig. R7 (g), (h) in green colour, is enforced by the glide mirror plane that is along ΓZ and perpendicular to Z-B direction. Notice that the nodal line Z-B can be extended to the KNL Z-L and part of the KNL Z-L are folded back on the Brillouin zone boundary as B₁-L (see Fig. R7 (g)).

Furthermore, another complication introduced by the nonsymmorphic operations (glide mirrors or screw rotations) as we discussed is that the nonsymmorphic operations can complicate the algebra and introduce higher-fold degeneracies at TRIMs. In the revised manuscript, we summarised symmetry-allowed higher-dimensional coreps at TRIMs for non-magnetic non-centrosymmetric achiral crystals in Table S5. It can be seen that most of these high coreps appear in nonsymmorphic SGs. For achiral noncentrosymmetric space groups, the appearance of high coreps at a TRIM can enforce several KNLs to touch at this TRIM. Through the materials examples with nonsymmorphic symmetries we gave in Fig. R7, it can be seen despite the complications we mentioned above, there are still KNLs coming from TRIMs with achiral symmetries (which contain operations $\{R|t\}$ with R being mirror or other roto-inversion symmetries).

Fig. R7 (a), (b) and (c), respectively, show the crystal structure, the Brillouin zone and band structure of TePb_2O_5 (SG No. 9, Bb C_{1v}). The KNLs in (b) are highlighted as the blue colour. (d) shows the energy difference of the two selected bands (red colour in (b)) at the glide mirror plane, where the positions of TRIMs are highlighted with dashed circles and the KNL ΓY is clearly visible. (e) shows the energy difference of the selected two bands with respect to the momentum k at the Brillouin zone boundary, and the KNL ZL_1 is perpendicular to the glide mirror plane. (f), (g) and (h), respectively, shows the crystal, the Brillouin zone and band structure of CsPbF_3 (SG No. 161, $R3c$ C_{3v}). The KNL enforced by the glide mirror at the Brillouin zone boundary is highlighted with green colour in (g) and (h).

In the revised manuscript, we added the discussions that the glide mirror symmetry can enforce KNLs at Brillouin zone boundary in the Supplementary Material Section 7B. And in Section 7F of the Supplementary Material, we present the allowed higher-dimensional coreps at TRIMs and more material analysis of materials with nonsymmorphic symmetries. However, a complete list of how the KNLs are connected in nonsymmorphic achiral crystals deserves more work in the future.

20. Throughout the SM, it should briefly be noted that the appearance of Kramers-Weyl points at high-symmetry points in achiral crystals was introduced in Ref. 55.

[Authors' response]

In the revised manuscript, we have pointed this out in the Section 5 of the Supplementary Material as

“As discussed in the main text, these chiral TRIMs that host electronic states described by a two-dimensional irreducible coreps of their little groups will emerge as Kramers Weyl points in achiral crystals. Notably, the appearance of Kramers Weyl points at high-symmetry points in achiral crystals was introduced in Ref. [S7] as well” (Ref. S7 is Nature Materials 17, 978–985 (2018)).

REVIEWERS' COMMENTS

Reviewer #1 (Remarks to the Author):

I have read the reply to the review report. I think that the reviewer did a great job in addressing questions from both reviewers. I highly recommend the paper for publication.

Reviewer #2 (Remarks to the Author):

The authors have made a strong effort to address all of my comments, and I believe that the presentation of this work has been significantly improved, and now meets the level required for publication in Nature Communications. I have a couple of very minor comments, which are listed below - after these comments are implemented, I recommend that this work be accepted for publication in Nature Communications.

Minor comments:

1. In the main text and in the SM, the authors note that the appearance of 8D coreps in achiral space groups (SGs) is consistent with "Bradlyn, etc." There are two issues with this sentence - first, stylistically, "etc." should be substituted with "et al." In general, etc. is a bit too informal to use in scientific writing, and is also incorrect in this instance. Second, 8D coreps were first recognized to appear in solid-state materials in Wieder, et al. (Ref. 23 in the revised manuscript) before the appearance of the more complete work by Bradlyn. Throughout this work, 8-fold degeneracies in 3D crystals should be attributed at least equally to both works.
2. In the same sentence in the main text and SM, the SG numbers, but not symbols, for SGs 218 and 220 are listed. The authors should also list the symbols for these SGs here, and should confirm that there are no other remaining instances in which SG numbers appear without the accompanying symbols.
3. In SM 5 B, the authors state that the "... little group of each TRIM can be identified by consulting the program KVEC on the Bilbao Crystallographic Server..." However, this statement is only partially correct -- KVEC is an older program that only incorporates the unitary subgroup of each little group. In nonmagnetic crystals, antiunitary (time-reversal) symmetry also plays a role. To incorporate the role of antinunitary symmetries in identifying little groups using the Bilbao server, one must combine the output of KVEC with the newer program "MKVEC" that was introduced in [Elcoro, et al., arXiv:2010.00598 (2020)] and [Y. Xu, et al., Nature 586, 702 (2020)].
4. In SM 7F, the word "symmorphic" is misspelled as "symmporphic."

Reviewer #1:

[General comments] I have read the reply to the review report. I think that the reviewer did a great job in addressing questions from both reviewers. I highly recommend the paper for publication.

[Authors' response]

We sincerely thank the reviewer for his/her strong support for the publication of this work.

Reviewer #2:

[General comments] The authors have made a strong effort to address all of my comments, and I believe that the presentation of this work has been significantly improved, and now meets the level required for publication in Nature Communications. I have a couple of very minor comments, which are listed below - after these comments are implemented, I recommend that this work be accepted for publication in Nature Communications.

[Authors' response]

We sincerely thank the reviewer for his/her careful reading of the manuscript and his/her support for the publication of this work. In the following, we address the reviewer's remaining minor comments point to point.

Minor comments:

1. In the main text and in the SM, the authors note that the appearance of 8D coreps in achiral space groups (SGs) is consistent with "Bradlyn, etc." There are two issues with this sentence - first, stylistically, "etc." should be substituted with "et al." In general, etc. is a bit too informal to use in scientific writing, and is also incorrect in this instance. Second, 8D coreps were first recognized to appear in solid-state materials in Wieder, et al. (Ref. 23 in the revised manuscript) before the appearance of the more complete work by Bradlyn. Throughout this work, 8-fold degeneracies in 3D crystals should be attributed at least equally to both works.

[Authors' response]

We thank the reviewer for noting this. We have changed "etc." into "et al.". We also agree with the reviewer that Wieder, et al. (Ref. 23) also recognized 8D coreps can be supported in achiral space groups.

To acknowledge this, we follow the reviewer's suggestion and changed "Bradlyn et al. [56]" as "**Wieder et al. [23] and Bradlyn et al. [56]**" in the revised main text and Supplementary information.

2. In the same sentence in the main text and SM, the SG numbers, but not symbols, for SGs 218 and 220 are listed. The authors should also list the symbols for these SGs here, and should confirm that there are no other remaining instances in which SG numbers appear without the accompanying symbols.

[Authors' response]

We thank the reviewer's suggestion on the presentations of SG symbols. In the revised manuscript, we have added the corresponding SG symbols for all SG numbers accordingly.

3. In SM 5 B, the authors state that the "... little group of each TRIM can be identified by consulting the program KVEC on the Bilbao Crystallographic Server..." However, this statement is only partially correct -- KVEC is an older program that only incorporates the unitary subgroup of each little group. In nonmagnetic crystals, antiunitary (time-reversal) symmetry also plays a role. To incorporate the role of antinunitary symmetries in identifying little groups using the Bilbao server, one must combine the output of KVEC with the newer program "MKVEC" that was introduced in [Elcoro, et al., arXiv:2010.00598 (2020)] and [Y. Xu, et al., Nature 586, 702 (2020)].

[Authors' response]

We thank the reviewer's suggestion. The references in [Y. Xu, et al., Nature 586, 702 (2020)] and [Elcoro, et al., arXiv:2010.00598 (2020)] are added the in revised Supplementary material as ref. [18], ref.[19], respectively.

4. In SM 7F, the word "symmorphic" is misspelled as "symmporphic."

[Authors' response]

We have corrected this typo in the revised manuscript.